# Analysis of Livestock Mobility and Implications for the Risk of Foot-and-Mouth Disease Virus Spread in Iran

Kamran Mirzaie [1,*], Shahir Mowlaei [2], Elena Arsevska [3], Bouda Vosough Ahmadi [4], Francesca Ambrosini [4], Fabrizio Rosso [4] and Etienne Chevanne [4]

[1] Animal and Animal Products-Borne Diseases Research Center, Iran Veterinary Organization (IVO), Tehran 1431683765, Iran
[2] Independent Researcher, Torbat-e Jam 95717, Iran
[3] UMR ASTRE (Unit for Animals, Health, Territories, Risks and Ecosystems), French Agricultural Research for International Development (CIRAD), 34398 Montpellier, France
[4] European Commission for the Control of Foot-and-Mouth Disease, Food and Agriculture Organization of the United Nations (FAO), 00153 Rome, Italy; bouda.ahmadi@fao.org (B.V.A.)
[*] Correspondence: dr.mirzaiee@gmail.com

**Abstract:** Foot-and-mouth disease (FMD) is endemic in Iran and associated with a large impact on the livestock industry. Livestock mobility is recognized as one of the most important risk factors for FMD and other infectious livestock diseases' introduction and dissemination. The description of temporal and spatial aspects of livestock mobility networks in Iran provides insights into FMD epidemiology. It also assists in the formulation of recommendations to mitigate the risk of animal disease transmission through livestock movements. This study is the first spatiotemporal description of official/registered cattle, sheep, goat, and camel movements in Iran, using records related to the period from March 2020 to August 2021 extracted from the Iran Veterinary Organization (IVO) Quarantine system. It shows that the static networks drawn by the movements of small ruminants, cattle, and camels to farms or slaughterhouses are complex and highlights the predominance of a few provinces and towns. In particular, the results show that Razavi Khorasan and West Azerbaijan provinces, major provinces for the Iranian livestock sector, are provinces where significant volumes of small ruminants and cattle are moved (from, to, or within) and, therefore, should be prioritized for targeted and timely risk reduction interventions. This study produces some of the necessary inputs for the risk assessment of FMD and similar transboundary animal diseases (TADs) spread within Iran, which is needed to regularly update the national risk-based control strategy for FMD and other TADs.

**Keywords:** livestock movement; network analysis; foot-and-mouth disease; progressive control pathway; Iran

## 1. Introduction

Foot-and-mouth disease (FMD) is a contagious infectious disease of cloven-hoofed animals caused by a picornavirus that belongs to the genus *Aphtovirus*, with seven known antigenically distinct serotypes. The disease is endemic in Asia, most sub-Saharan Africa, and the Middle East [1]. The disease causes direct losses of livestock production, and its occurrences impact the international trade of livestock and livestock products.

The circulation of FMD viruses (FMDV) seems to occur within the "pools", defined as regions characterized by the predominance of certain FMDV serotypes and strains [2]. The Islamic Republic of Iran (further referred to as Iran) belongs to Pool 3, i.e., the West Eurasia pool, where FMDV serotypes O, A, and Asia 1 are historically circulating [1,2]. The FMDVs can spread from their original pool to another one, with the potential to cause significant outbreaks in livestock not previously exposed to the virus serotype. Such an incursion would lead to a reevaluation of the control strategy and the selection of appropriate vaccines.

For instance, in late 2022, the incursion of an exotic SAT2 serotype in West Eurasia from Pool 4, "Eastern Africa", illustrated the FMDV dynamics between pools and highlighted issues of global FMD vaccine security [3].

Iran is a vast country sharing borders with countries showing different animal health statuses, making animal disease management a complex task for the livestock industry and veterinary services.

In Iran, FMD is endemic and occurs throughout the year, with the highest incidence observed between January and March [4]. A recent study investigated socio-environmental determinants of FMD incidence in Iran at the provincial level based on 2016–2017 surveillance data; the highest FMD incidence (number of cases per 1,000,000 animals) was observed in Razavi Khorasan and Fars provinces [5], located in Northeastern and Southwestern Iran, respectively. The number of FMD outbreaks, based on clinical findings from passive surveillance, is high but still considered underestimated [6]. A serosurvey conducted in 2015 for detecting antibodies against nonstructural proteins of FMDV among young calves in West Azerbaijan (a cattle-dense province located in the Northwestern region of Iran) revealed a very high level of FMD infection in villages, dairy, and beef farms [7]. Based on surveillance data between January 2017 and March 2019, the highest proportion of outbreaks was recorded in villages (75%); industrial and semi-industrial farms accounted for 18% of all reported outbreaks [4].

The importance of Iran as a site for generating genomic diversity and a gateway for South Asian FMDV for Europe was suggested in a recent study [8]. The westward movement of FMDV subtypes through Iran can threaten neighboring countries and, eventually, the European continent, which is currently FMD-free. Trade and associated movements of livestock play a fundamental role in the transboundary spread of infectious diseases such as FMD, bovine tuberculosis, Johne's disease, or brucellosis [9,10]. This has also been the case in Iran, where one study carried out in West Azerbaijan province found that the odds of being infected with FMDV were higher among calves that came from farmers involved in livestock trading compared to livestock owned by farmers not engaged in trading [7]. Furthermore, in the Razavi Khorasan province, livestock transportation was identified as an important risk factor for farm-level FMD infection [11].

In Iran, livestock production and trade play a crucial role in improving the livelihoods of people and are expected to play an even greater role as the human population and demand for animal protein increase. For example, the production of meat (cattle, sheep, and goat meat) increased from 327,000 tons to 884,000 tons between 1974 and 2020 (average annual growth rate of 2.19); milk production increased from 2,000,000 tons to 11,000,000 tons between 1974 and 2020 (average annual growth rate of 3.51). In Iran, cattle and small ruminant farming is practiced throughout the country [5]; 90% of small ruminants and 65% of cattle are bred by smallholder farmers in rural areas (villages) and in small mixed herds (cattle, sheep, and goats). Camel breeding is mainly performed in the desert and semi-desert areas in Iran, thus ensuring self-sufficiency in meat production for the local population [12]. There is an intense trade of live animals and animal products between actors of the livestock value chain (farmers and dealers/traders, in particular) and between villages, towns, and provinces within Iran. In some countries, the Muslim holiday *Feast of Sacrifice*, *Eid al-Adha* in Arabic and *Tabaski* in Wolof, has been associated with intense livestock mobility [13]. However, little is known about the importance and spatiotemporal patterns of livestock mobility in Iran.

In recent years, there has been increased consideration of FMD control in endemic countries to improve livestock productivity and livelihoods and protect FMD-free countries or zones from the risk of (re-)introducing the virus. The Global FMD Control Strategy endorsed by the Food and Agriculture Organization of the United Nations (FAO) and the World Organisation for Animal Health (WOAH, founded as OIE) in June 2012, aims to foster the application of the progressive control pathway for FMD (PCP-FMD) in endemic settings through the development and implementation of national FMD control programs, supported by local economic drivers for public and private investment. The Iran Veterinary

Organization (IVO) is currently implementing a Risk-Based Strategic Plan (RBSP) that emphasizes the need to carry out regular risk assessments for FMD introduction and spread in key livestock value chains.

In this context, this study aims to provide the first spatiotemporal description of livestock mobility networks in Iran. It is envisaged that these findings will contribute to the risk assessment of FMD and other TADs spread in Iran and the region overall, as livestock mobility is considered a key risk factor for FMDV spread.

## 2. Materials and Methods

### 2.1. Dataset

The data were obtained from IVO's quarantine system and cover the records of official legal livestock movements within Iran (not including cross-border movements) for farming (breeding or fattening) and slaughter between 20 March 2020 and 7 August 2021. When livestock owners move their animals within Iran, they must contact the veterinary services at the town level to record the movement in the quarantine system. This record will then be verified by the veterinary administration at the destination. It can refer to one or several animals (a batch). Transhumance movements are also recorded this way.

For the purpose of this study, we considered the source and destination towns, origin type (*farm* or permanent *livestock market*), destination type (*farm* or legally registered *slaughterhouse*), shipment date (date of departure), headcount, and the species of the animals transferred. The latter attribute covers *small ruminants* (sheep and goats), *cattle*, *camels*, *buffalos*, and *other* animals (dogs, cats, wild animals, and animals used in laboratory experiments). In the remainder of this study, we discard *buffalos* (too few records, namely, 102) and *other* animals (1759 records), as the latter category is considered insignificant in FMDV transmission. Accordingly, and unless stated otherwise, the label *all*, representing all species, will refer to the first three, namely, small ruminants, cattle, and camels. Movement records occurring within the same town (same origin and destination town) were removed from the current analysis. Origin and destination types are attributes of the record and not inherent to the former, so a town may assume different types/roles in different records. This is due to the fact that, generally, a town represents multiple epidemiological units.

### 2.2. Network Construction

2.2.1. Aggregation Models

For each group of species—all, small ruminants, cattle, and camels—and for the ternary choice of whether to include slaughterhouses with farm destinations or not or include only them, one can construct three basic static network models (unweighted, frequency, and volume model) of the data [14]. With regard to the temporal view of the data in the following analyses, each time slice represents a static network constructed with a specific choice of one of these basic models, with records restricted to the corresponding time interval of the given time slice.

In the frequency model, with each ordered pair of origin and destination towns, one associates a directed edge with a weight equal to the number of corresponding records in the dataset. The edge weights, therefore, encode the frequency of the directed relation (i.e., shipment) between origin and destination towns with regard to the time span of the dataset.

In the volume model, the edge weights correspond to the total sum of the headcounts transferred between every ordered pair of origin and destination towns. These models, frequency, and volume, correspond to different attribute aggregation schemes, and, as such, they are the simplest edge-weighting strategies in network construction.

2.2.2. Dissimilarity Inference

The weights attributed to the directed edges of the constructed network models are perceived as degrees of the (directed/ordered) similarity. The similarity interpretation is suitable for matrix-based and other graph metrics that interpret edge weights as a measure

of (oriented/ordered) proximity and nonexistent edges (in either direction), implicitly assuming sufficiently small weights (typically zero [15,16]). This is the motivation behind the (nontrivial) aggregation strategies of the above network construction models. For distance-based metrics, however, the weight of the connection between an (ordered) pair of nodes is a measure of distance (difference), and nonexistent edges implicitly assume sufficiently large weights (typically positive infinity).

In some undirected network (construction) models, the similarity between pairs of nodes is inferred from that of certain node characteristics. In such cases, the inference of dissimilarity may be reduced to a shift in the former's tone. As our similarity network models are, instead, inferred from characteristics of the ordered relation on pairs of nodes, the desired dissimilarity between them must be fabricated by other means. Of the various conceivable strategies, we employ weight reversal: edges with a higher similarity weight are attributed the lower as their dissimilarity weight.

### 2.2.3. Orientation Reversal

For certain matrix-based centrality measures, such as hub and authority scores [17], given that the edges of the present networks encode shipment/supply, as opposed to demand, we reverse the orientation of edges. In the special case of hub and authority scores, this reversal is equivalent to the transposition of the network's adjacency matrix, which, in turn, amounts to the node-wise exchange of these scores. In general, the reversal allows us to utilize such measures in the proper orientation (direction of edges) that yields meaningful results.

### *2.3. Static Measures*

### 2.3.1. Global Clustering Coefficient

The global clustering coefficient is a normalized measure of the frequency of triangles in paths of the length of two of the undirected unweighted networks. It is a measure of the degree of transitivity of relations in the network. Among variants, we employ the following weighted definition:

$$gcc(\cdot) \equiv \frac{\mathbf{Tr}A^3}{\sum_{i \neq j} A_{ij}^2} \tag{1}$$

where $A$ is the undirected adjacency matrix of the network with similarity weights and $\mathbf{Tr}(\cdot)$ is the trace operator.

### 2.3.2. Global Efficiency

Global efficiency is a measure of how close (ordered) pairs of nodes are in the network and is defined as

$$ge(\cdot) \equiv \frac{\sum_{i \neq j} d_{ij}^{-1}}{n \cdot (n-1)}, \tag{2}$$

where $d_{ij}$ is the distance of node $i$ from node $j$, and $n$ is the number of nodes of the network. Higher values of global efficiency correspond to lower topological resistance, measured in shortest path length, against nodes transitively influencing each other. This, however, is a generic characteristic of global measures of connectedness.

### 2.3.3. Local Efficiencies

For directed networks, local efficiency is a restriction of the definition of global efficiency, Equation (2), to a fixed index, but now, one is faced with a choice. Fixing the first index, $i$, corresponds to the efficiency of a node in transitively reaching other nodes through directed paths, what may be called the forward efficiency of the node $i$. Similarly, fixing the second index, $j$, yields the backward efficiency of the node $j$.

### 2.4. Hub and Authority Scores

In directed networks, the hub and authority centralities are middle-shifted degree-weighted versions of the notions of source and sink, respectively, by conditioning them on each other [17]. We used the hub and authority scores to estimate the relative importance of certain towns in the network of animal mobility in Iran. In the context of infectious disease spread, towns with high hub centrality have a high amount of outgoing movements and can indicate locations where the virus can more easily and more quickly spread. Towns with high authority centrality have a relatively high amount of incoming traffic, making them particularly susceptible to pathogen introduction [18].

In the following, we utilize it as an auxiliary measure to reweight our modified forward and backward efficiency scores. In applying the Hyperlink-Induced Topic Search (HITS) algorithm to compute the hub and authority scores, we reverse the orientation of edges to correctly account for the semantic content of this attribute (shipment/supply rather than referral/demand). As previously mentioned, this is equivalent to transposing the adjacency matrix and amounts to the exchange of the corresponding hub and authority scores.

### 2.5. Hybrid Centralities

The ranking of nodes obtained from different centrality measures can be combined in various ways. Here, we employ a simple biased (star-like) strategy that treats one of the rankings as primary and all other as secondary. In general, this is to reflect one's estimate of the utility/role of these rankings in portraying the centrality measure of which the composed expression is a proxy; see the following topic. For the base case of two rankings $a$ and $b$, with $a$ primary and $b$ secondary, we define the new ranking of a node $v$ as

$$a_v \cdot (1 + \mathrm{Fr}_v(b)) \tag{3}$$

where $\mathrm{Fr}_v(b)$ is the fraction of nodes that rank (score) lower than $v$. For multiple secondaries, one can proportionately extend the sum, possibly accompanied by nontrivial relative weights. In the context of infectious disease spread, the hybrid centralities allow us to identify influential nodes in the animal mobility network in Iran.

### 2.6. Prevention versus Detection

Consider the qualitative notions of generalized sources and sinks in directed networks. These are nodes that, through (shorter) directed paths that, in the case of the frequency network model, represent shorter temporal (or equivalently more probable) successions, can reach or be reached by a large fraction of nodes of the network, respectively. The duality between such (generalized) sources and sinks mirrors the duality between nodes most suitable for targeted disease prevention measures and disease detection (sentinel placement) under limited resources, respectively.

This is justified by the observation that higher-ranking hot sources (sinks) are nodes that (are) influence (d by) a large portion of the network in relatively shorter spans of time (or through higher percolation probabilities) and, as such, prioritized maintenance (monitoring) of their health status under limited budget is a sensible prevention (detection) strategy.

The above distance-based forward and backward efficiencies constitute one such example of paired network measures that may be employed. However, these measures are not sensitive to the importance of individual nodes (in their sums over all nodes). Therefore, a natural way to extend forward and backward efficiencies is to weigh their corresponding sums with the relative importance of the node to and from which the distance is computed with respect to the specific measure. The latter means that, for forward efficiency, for instance, the accompanying weight encapsulates the ability of the target node to facilitate/improve the reachability of farther nodes for the source node whose weight efficiency one is interested in. One such choice of weight that is insensitive to the choice of measure, and so it can also be used for weighting backward efficiency, is the betweenness of the (target/source) node. A measure-specific choice, on the other

hand, is the weighted outdegree, which can be seen as the product of the mean of the weight of out-edges of the node and its unweighted outdegree. To suppress the impact of outliers, however, we replace the mean statistics with the median and further replace the weight of out-edges with the form in Equation (3), where $b$ is the median-weighted outdegree of the target node of the given edge, that is, the product of the median of the weights of out-edges of the latter node, excluding the node $i$, and its unweighted outdegree. As a result, not only the weight of the outgoing edge of the target node (whose distance from the primary/source node of the measure is computed) but also the (normalized) median-weighted outdegree of the target node of this edge is (secondarily) accounted for. Figure 1 illustrates this configuration.

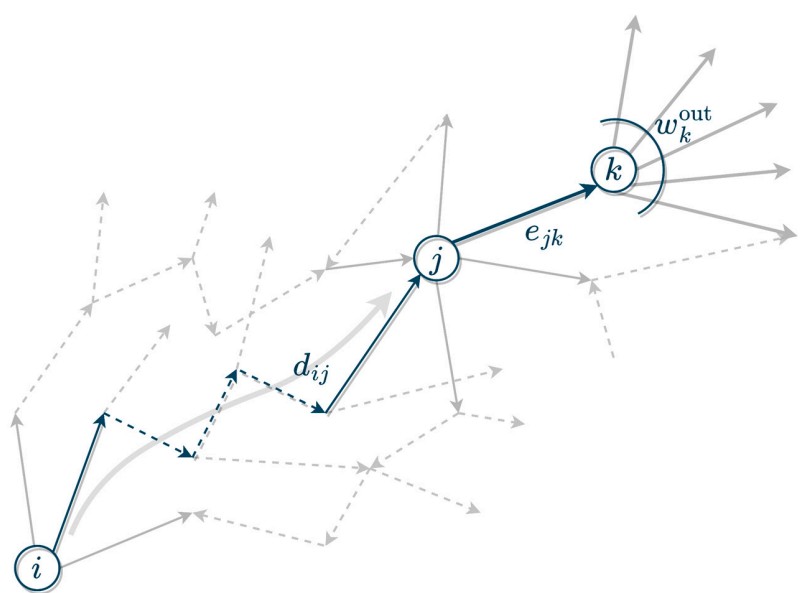

**Figure 1.** The configuration pertaining to the expression of the weighted forward efficiency (source-ness) of the node $i$ in Equation (4). $e_{jk}$ is the weight of edge $(j, k)$, and $w_k^{\text{out}}$ is the median-weighted outdegree of node $k$, excluding the node $i$.

So far, for every node $j \neq i$ to which the node $i$ has a finite distance (infinite distance indicates the lack of a directed path), we have computed a score that reflects its capacity to facilitate the shortest paths from the node $i$ to other nodes of the network. This score, normalized, is then used to transform the inverse of the distance from $i$ to $j$ by further linearly shifting its original value toward the median of its support over all nodes $\neq i$; the lower the score is for inverse distances greater than the median, the higher the score for those that are not. The resulting expression, which we will refer to as the *source*(*ness*) score of the node $i$, takes the following form:

$$\text{source}(i) \equiv \sum_{j \neq i} f\left( d_{ij}^{-1}; \, \underset{k \neq i}{\text{median}}\left( e_{jk} \cdot \left(1 + \text{Fr}_k\left(w^{\text{out}}\right)\right)\right) \cdot \left| \left\{ k \neq i : e_{jk} \neq 0 \right\} \right| \right) \quad (4)$$

where $e_{jk}$ is the weight of the outgoing edge from node $j$ to node $k$; $w_k^{\text{out}}$ is the median-weighted outdegree of the latter, excluding the node $i$, and the function $f$ encodes the above shift toward the median of inverse distances. Similarly, from backward efficiency, we can extrapolate the *sink*(*ness*) score of the node $i$ as

$$\text{sink}(i) \equiv \sum_{j \neq i} f\left( d_{ji}^{-1}; \, \underset{k \neq i}{\text{median}}\left( e_{kj} \cdot \left(1 + \text{Fr}_k\left(w^{\text{in}}\right)\right)\right) \cdot \left| \left\{ k \neq i : e_{kj} \neq 0 \right\} \right| \right) \quad (5)$$

where $e_{kj}$ is the weight of the incoming edge, and $w_k^{\text{in}}$ the median-weighted indegree, excluding the node $i$.

Expressions (4) and (5) are intended to emphasize the ease (higher probability) of (indirect) access to and from other nodes of the network, respectively. We utilize these measures to rank the nodes of the static frequency networks, where high-ranking sources are considered suitable candidates for targeted preventive measures and, dually, high-ranking sinks are deemed suited for targeted detection/surveillance. As previously mentioned, these measures are further augmented by their corresponding authority and hub scores via Equation (3), and the hybrid scores reported in Section 3 correspond to this adjustment.

In a directed network, we say that the node *i* is *in-closer* (*out-closer*) to the node *k* than the node *j* if $d_{ki} < d_{kj}(d_{ik} < d_{jk})$. Nodes that are in/out-closer to top prevention/detection candidates, *likely* authorities/hubs, are, in turn, better candidates for detection/prevention investment. However, under constrained resources, one would want to choose those nodes that are in/out-close to as many such nodes as well as to the rest of the nodes in the network. It is our estimate of the above likelihood against the latter property desired that drives our choice of a star composition, as in Equation (3), over direct multiplication of source and sink scores by their authority and hub counterparts.

*2.7. Temporal Analysis*

The common approaches to temporal network analysis may be realized as three types. In time-sliced analysis, one studies the evolution of certain aspects (measures) of a temporal network by estimating them on static networks that are obtained by aggregating over time slices (windows) of its temporal span. Aggregation is carried out by the previously discussed network construction models, and the result is a sequence of estimates of the measure of interest across distinct time windows.

With a temporal measure, one may also simply retain the (higher) native temporal resolution to examine the evolution of the given measure.

A third, holistic approach utilizes the entire time span of the network to generate estimates of its various global and local aspects. Some of these estimates are attempts at generalizing existing static measures by means of kernels or otherwise; others tilt static structural constructs along the temporal dimension with new meanings and constraints [19].

In this study, given the resources, we decided to adapt the first approach and use 11-day time slices for about 1.5 weeks to capture finer aspects of the evolution of measures.

*2.8. Spatial Analysis*

We aggregate the data at two different levels.

The town-level resolution (where a node of the networks corresponds to a town) is considered in the analysis to account for key urban consumption centers and specific premises and may be useful for IVO to target specific resource-demanding activities with provincial veterinary administrations, such as active surveillance efforts. It is worth noting that as a node can host a farm-type destination and/or a slaughterhouse-type destination, the total number of nodes in the following description does not correspond to the sum of nodes to farms only and to slaughterhouses only.

Data is also aggregated (summed up) at the provincial level, as it represents the spatial resolution relevant to decision-making in disease control strategies. The central veterinary services (IVO) implement control measures such as vaccination campaigns at the provincial level in case of outbreaks. In particular, strip plots are used to visualize the range and dispersion of records at the provincial level, distinguishing movements within a province, out of a province, or into a province. Figures are drawn considering the species involved and the destination type (farm or slaughterhouse).

The analyses and data visualization were performed using the Python programming language [20].

*2.9. Lateral Trajectories*

We study the latitudinal and longitudinal distribution of shipments, as the respective differences in those associated with their respective destinations and origins, in the network

as an indication of its capacity for the spread of disease over larger spatial scales within relatively shorter periods of time.

## 3. Results

Post elimination, the dataset contains 185,967 movement records, and 99.1% of these records describe animal movements between distinct origin and destination towns (Figure 2; see Supplementary Figure S5 for the spatial distribution of farms, markets, and slaughterhouses, aggregated at the village level).

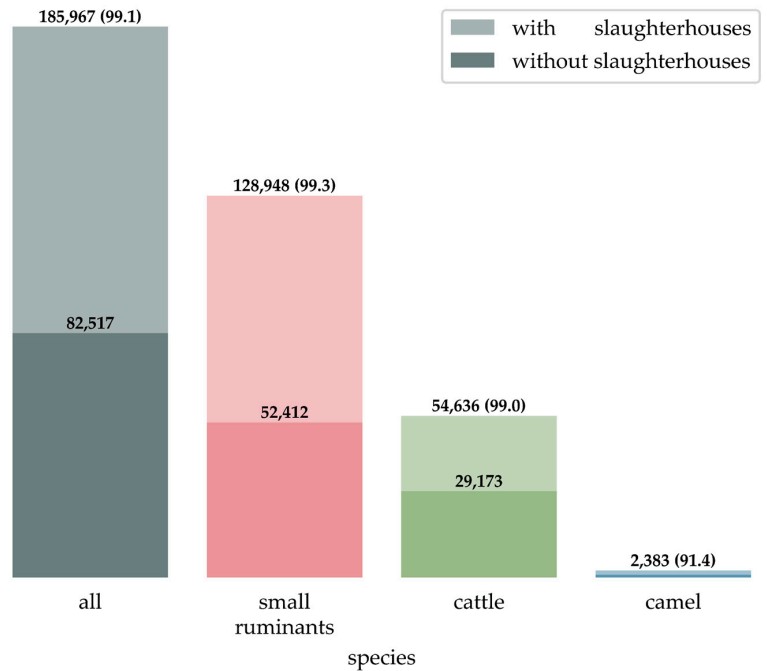

**Figure 2.** Overview of the animal mobility dataset, indicating the proportion of records by animal species and destination to slaughterhouses. One record refers to the movement of one or several animals. Records of movement with identical origin and destination towns (1679 in number) were discarded from the current analysis, and the number in parenthesis represents the percentage of records kept for further analysis. As an illustration: there are 54,636 movement records for cattle (corresponding to 99% of the total records for cattle; i.e., the 1% of cattle movement records were describing movements occurring within the same town and therefore discarded; 29,173 of the 54,636 movement records were directed to farms, the rest to slaughterhouses. Note: All pale colors are pertinent to statistics with slaughterhouses and other dark colors are related to statistics without slaughterhouses.

The overall network includes 418 nodes distributed in 31 out of 32 provinces in Iran and 185,967 edges, describing the movement of 9,767,689 animals (cattle, sheep, goats, and camels) from 20 March 2020 to 7 August 2021 (Table 1). At first glance, the small ruminant network includes a larger number of nodes (94.7% of all nodes) and edges (69.3% of all edges) and interests a larger volume of animals (93.9% of animals) than—in order—the cattle network and the camel one. For small ruminants, cattle, and camels, movement records (edges) mostly originate from farms rather than livestock markets (71.0% of all edges and 68.8% of animals); livestock markets are more frequently the origins of small ruminant movements compared to cattle and camel movements (35.5%, 14.6 and 3.2%, respectively). The vast majority of nodes of the small ruminant, cattle, and camel networks (96.5, 94.9, and 87.1%, respectively) include farm-type destinations. In small ruminant and camel networks, edges tend to be more directed to slaughterhouses (59.4 and 60.9%, respectively) than to farms, as opposed to the described cattle network. Volumes of small ruminants and camels moved to slaughterhouses are higher than to farms (61.3 and 56.7%, respectively), as opposed to cattle networks.

**Table 1.** Number of nodes (towns), edges, and animals moved within the static network during the study period. Percentages appear in bold within parentheses.

| | # Nodes (Towns) | | | # Edges (Frequency) | | | | | Headcount (Volume) | | | | |
| --- | --- | --- | --- | --- | --- | --- | --- | --- | --- | --- | --- | --- | --- |
| | All Nodes | Farm | Slaughter | All Edges | From Farm | From Market | To Farm | To Slaughter | All Heads | From Farm | From Market | To Farm | To Slaughter |
| All species [1] | 418 | 411 **(98.3)** | 233 **(55.7)** | 185,967 | 132,095 **(71.0)** | 53,872 **(29.0)** | 82,517 **(44.4)** | 103,450 **(55.6)** | 9,767,689 | 6,724,250 **(68.8)** | 3,043,439 **(31.2)** | 3,892,533 **(39.9)** | 5,875,156 **(60.1)** |
| Small ruminants | 396 **(94.7)** | 382 **(96.5)** | 211 **(53.3)** | 128,948 **(69.3)** | 83,109 **(64.5)** | 45,839 **(35.5)** | 52,412 **(40.6)** | 76,536 **(59.4)** | 9,172,492 **(93.9)** | 6,197,259 **(67.6)** | 2,975,233 **(32.4)** | 3,547,161 **(38.7)** | 5,625,331 **(61.3)** |
| Cattle | 352 **(84.2)** | 334 **(94.9)** | 172 **(48.9)** | 54,636 **(29.4)** | 46,679 **(85.4)** | 7957 **(14.6)** | 29,173 **(53.4)** | 25,463 **(46.6)** | 577,308 **(5.9)** | 509,318 **(88.2)** | 67,990 **(11.8)** | 337,633 **(58.5)** | 239,675 **(41.5)** |
| Camel | 116 **(27.8)** | 101 **(87.1)** | 60 **(51.7)** | 2383 **(1.3)** | 2307 **(96.8)** | 76 **(3.2)** | 932 **(39.1)** | 1451 **(60.9)** | 17,889 **(0.2)** | 17,673 **(98.8)** | 216 **(1.2)** | 7739 **(43.3)** | 10,150 **(56.7)** |

[1] "All species" here refers to small ruminants, cattle, and camels.

In Figure 3, towns are arranged in decreasing order of the total volume of animals moving to farms or to slaughterhouses for small ruminants, cattle, and camels. This graph is supplemented with information on the frequency of shipment to farms or slaughterhouses. Despite the high number of towns involved in the static networks (Table 1), some towns (less than 12 for small ruminants, fewer than 10 for cattle, and fewer than 5 for camels) tend to dominate the networks to farms and slaughterhouses. Frequency and volumes tend to be correlated. Small ruminant and cattle networks are dominated by the same towns: the town of Taybad (Razavi Khorasan province) dominates the small ruminant farming network, while Taybad and Orumiyeh (West Azerbaijan province) dominate the small ruminant slaughter network; Taybad and Orumiyeh dominate the cattle farming network, while the cattle slaughter network is again dominated by Taybad. The town of Birjand (South Khorasan province in Eastern Iran) dominates the camel farming network, while the town of Baft (Kerman province in Southeast Iran) dominates the camel slaughter network. In addition to the above-mentioned towns/provinces, Aligudarz (Lorestan province), Ahar (Zanjan province), and Esfaryen (North Khorasan province) are the towns/provinces that have the greatest volume/frequency of animals moved during the study period.

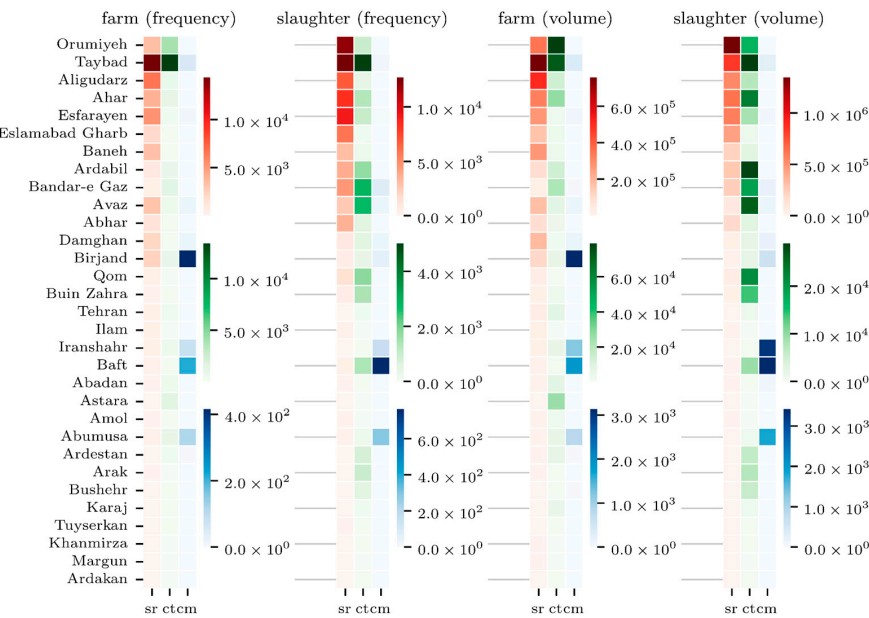

**Figure 3.** Total volume and frequency of export of small ruminants, cattle, and camels by town and both types of destination (farm and slaughterhouse).

### 3.1. Temporal Description

During the time span of the dataset, the small ruminant mobility largely outnumbered that of cattle and camels both in terms of the number of records (Figure 4a) and the volumes of animals moved (Figure 4b); small ruminant shipments are primarily directed to slaughterhouses. Movements of small ruminants, and to a lesser extent cattle, tend to increase during the study period.

Peaks in the volume of small ruminants moved to slaughterhouses were reached on two occasions during the study period, on 31 July 2020 and 21 July 2021, corresponding to Tabaski celebrations during which small ruminants (sheep and goats) are sacrificed (Figure 4). Peaks in the volume of small ruminants moved to farms are also reached before Tabaski celebrations. Iranian nomads move their livestock (mostly small ruminants) to grazing lands from May to September, but seasonal migration patterns cannot be identified in Figure 4. However, our dataset covers only the 2020 transhumance and the impact of COVID-19 movement restrictions on pastoralists remained undocumented. A decline in the volume of animals moved around mid-March 2021 may be attributed to the Persian New Year, called Nowruz, celebrated on 21 March 2021. As our data cover only one such occasion, a trend cannot be conjectured.

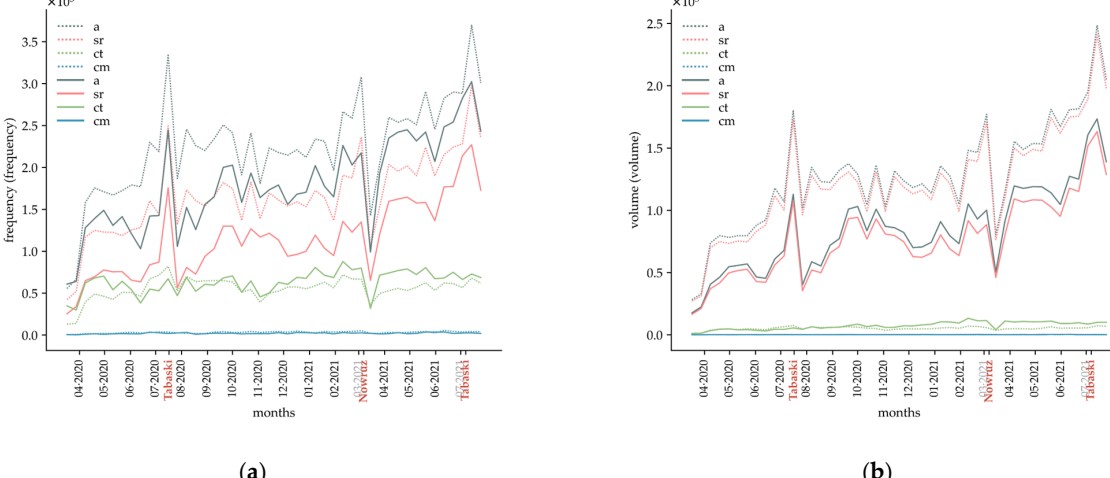

(**a**)        (**b**)

**Figure 4.** Monthly variations in movement records in Iran over the study period. (**a**) Variations in the frequency of records (using an 11-day time window). (**b**) Variations in the volume of animals moved (using an 11-day time window).

In these graphs, "a" refers to all species; "sr" refers to small ruminants; "ct" refers to cattle, and "cm" refers to camels. An 11-day time window is used for the time-sliced analyses to track the evolution of network measures across the time span of data. Labels of months on the *x*-axis mark the 15th day of each month. The timing of Tabaski and Nowruz celebrations in the study period is indicated on the *x*-axis. Dashed lines correspond to the records with slaughterhouses as the destinations, while solid lines correspond to the records with farms as the destinations.

To better illustrate the evolution of trade frequency and volumes of cattle and camels during the study period, a normalized form of the graph is shown in Figure 5. While the spike in trade volume around Tabaski seems to remain a characteristic of shipments of small ruminants, the decline around Nowruz is shared between shipments of small ruminants and cattle regardless of the destination type. Usually, in traditional camel breeding, animals are gathered in spring for shearing, treatment, identification through cauterization, and eventually sale before releasing camels in summer [12].

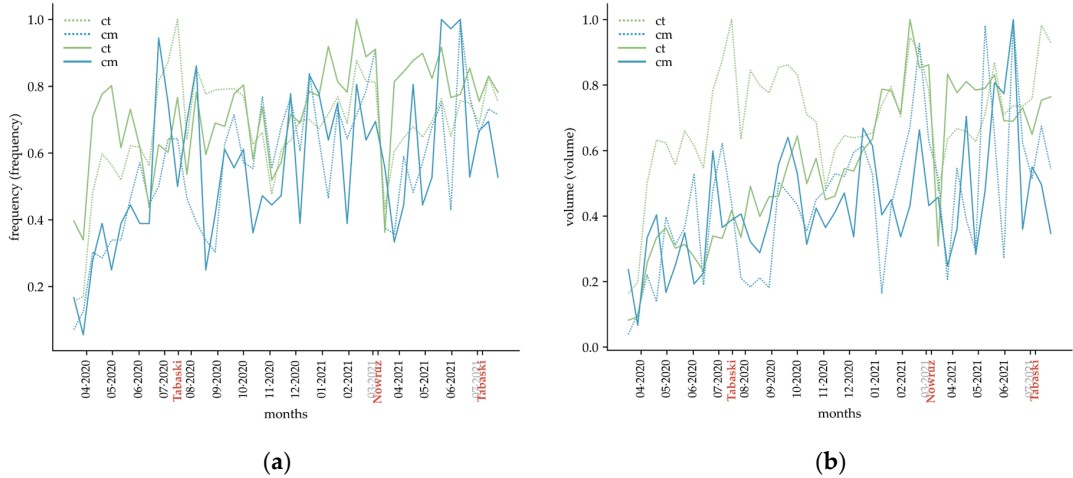

(**a**)        (**b**)

**Figure 5.** Normalized variations in the frequency (**a**) and volume (**b**) of cattle and camels moved over the study period (using an 11-day time window). In this graph, "ct" refers to cattle, and "cm" refers to camels. Dashed lines correspond to the records of slaughterhouses as the destinations, while solid lines correspond to the records of farms as the destinations.

Figure 6 depicts the time-sliced evolution of the global clustering coefficient (*gcc*) of the frequency (model of) network. The *gcc* values substantially vary during the study period; the highest peaks are visible in small ruminant networks (both for movements directed to slaughterhouses and farms) around Tabaski. This can be interpreted as nodes within the networks clustering more around these celebrations. Such a trend cannot be inferred from the graph for cattle and camel networks.

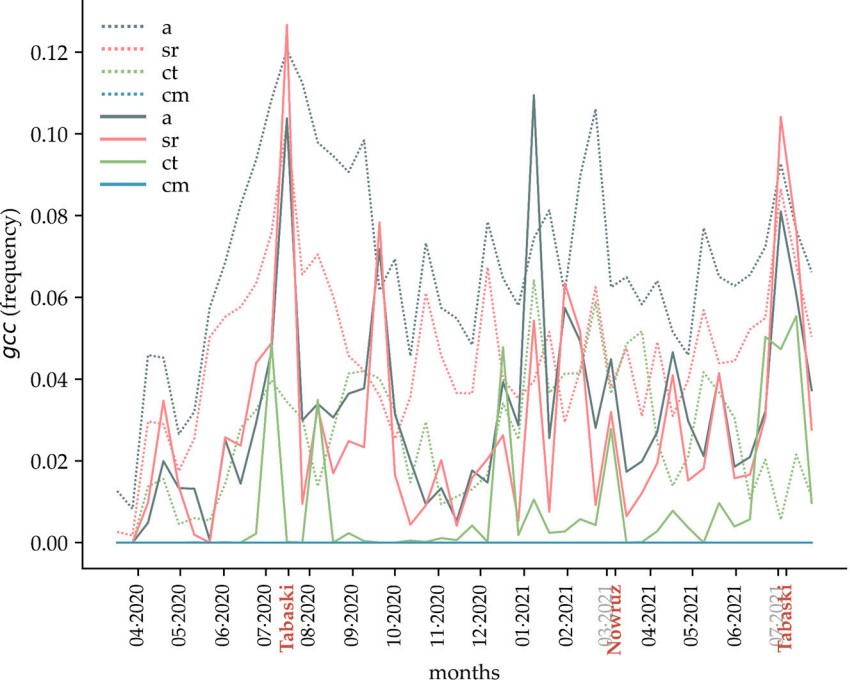

**Figure 6.** Variations in the global clustering coefficient (*gcc*) of the frequency networks over the study period. In this graph, "a" refers to all species, "sr" to small ruminants, "ct" to cattle, and "cm" to camels. Dashed lines correspond to the records with slaughterhouses as the destinations, while solid lines correspond to the records with farms as the destinations.

### *3.2. Spatial Description*

3.2.1. At Provincial Level

Small Ruminants

In the following strip plots (Figure 7a), provinces are arranged in the decreasing lexicographic order of, first, their (intra) export and, second, (intra) import frequencies and volumes for farming destinations. Import and export here refer to the movement into a province and out of a province, respectively, and do not correspond to the international trade of livestock. This order is maintained in the corresponding heatmaps (components (b) and (c) of Figure 7). The same approach is applied to Figures 8–12.

For ease of reference, in this work, we may abbreviate the province names *Chahar Mahall and Bakhtiari* to "C.M." and B.", *Kohgiluyeh and Buyer Ahmad* to "K. and B.A.", *Razavi Khorasan* to "R. Khorasan", and *Sistan and Baluchestan* to "S. and B".

Razavi Khorasan and Lorestan (in Western Iran) are the two top exporter provinces (in frequency and volumes) of small ruminants for farming, while Razavi Khorasan and Esfahan (in Central Iran) are the largest importer provinces (Figure 7a–c). R. Khorasan, Fars, and Semnan (in Northern Iran) are among the provinces showing three types (in, out, intra) of small ruminant movements for farming (Figure 7a). Razavi Khorasan, Lorestan, and Esfahan provinces seem to be characterized by the relatively small batches of small ruminants moved (mean volumes below 80 heads, irrespective of the type of movement). The mean volume of small ruminants moved to farms varies greatly between provinces and by type of movement (out, in, intra), with mean numbers ranging from 20 to 176 small

ruminants (Figure 7a, volumes in strip plots are capped at 200 heads). In Razavi Khorasan and, to a lesser extent, in Fars and Semnan provinces, there are frequent intra-provincial shipments of small ruminants (involving a large number of animals) for farming purposes (Figure 7b,c; Supplementary Figure S2).

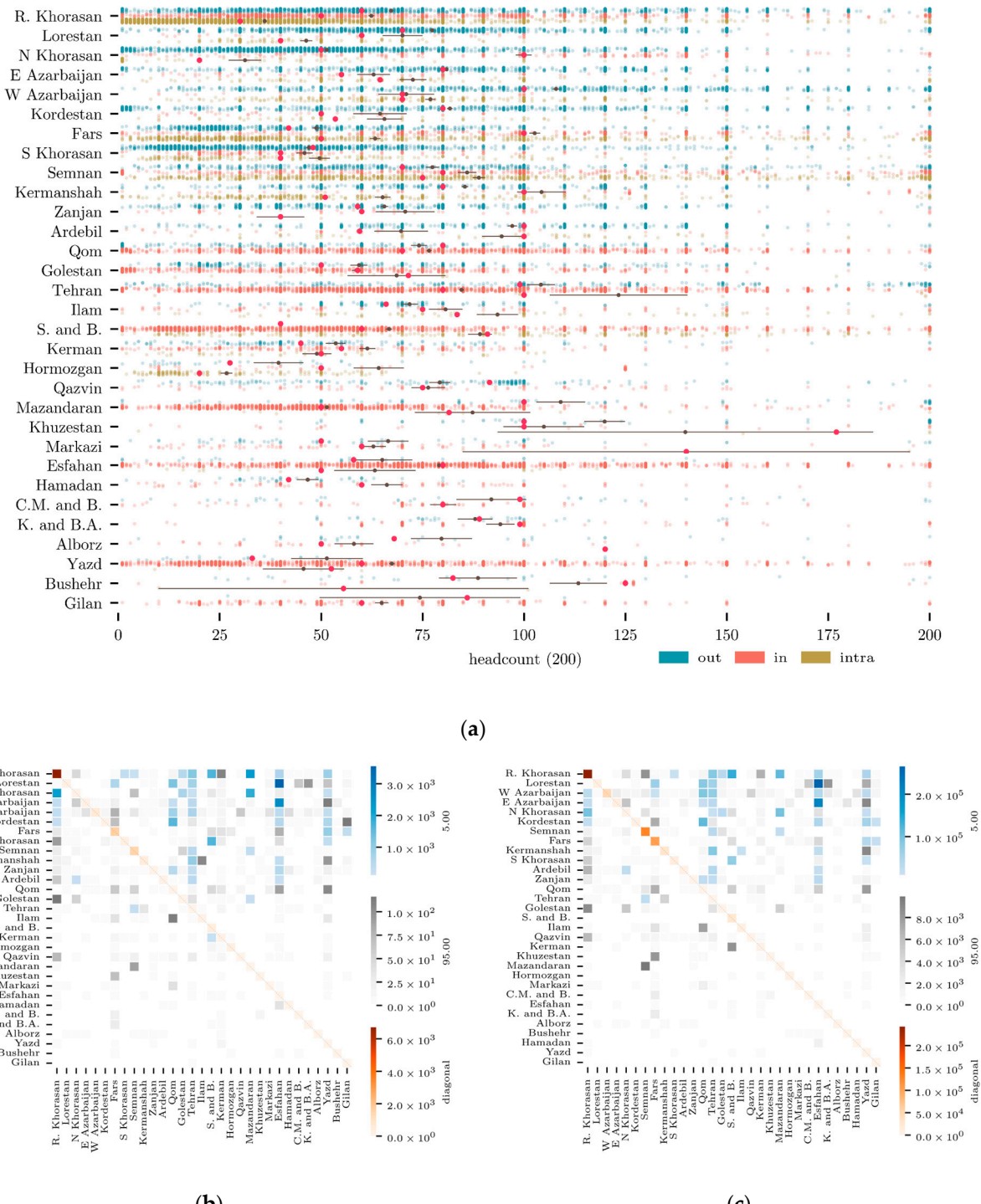

**Figure 7.** Small ruminant shipments in Iran for farming only. (**a**) Strip plot of shipments by province and headcount. On the *y*-axis, provinces are arranged in the decreasing order of, first, export and, then, import frequencies (including themselves). Shipments are further characterized as shipments to (in), from (out), and within (intra) the province. On the *x*-axis, volumes are capped at 200 heads of small ruminants, replacing larger quantities with this limit to aid visibility. Red points indicate the median,

and the darker-colored mean is accompanied by its corresponding standard error; they correspond to the uncapped data. In strip plots, adhering to the same order for frequencies, shipment volumes (headcounts) greater than the number in parentheses are set to this value. (**b**) Heatmap of the cumulative frequency of shipments between pairs of provinces. Rows and columns correspond to origin and destination provinces, respectively. The order follows that of (**b**). The initial choice of separation at 95% is recursively increased by halving its distance to 100% until the corresponding quantile is at least the other arbitrary number 10. Excluding the intra-type of shipments (diagonal, the red scale), the top 5% of the (remaining aggregated) shipments are expressed on the blue scale, while the rest are delegated to the grayscale. (**c**) Heatmap of the cumulative volume of small ruminants moved between pairs of provinces. The plot imitates Figure (**b**) to describe shipment volumes between provinces instead of frequencies.

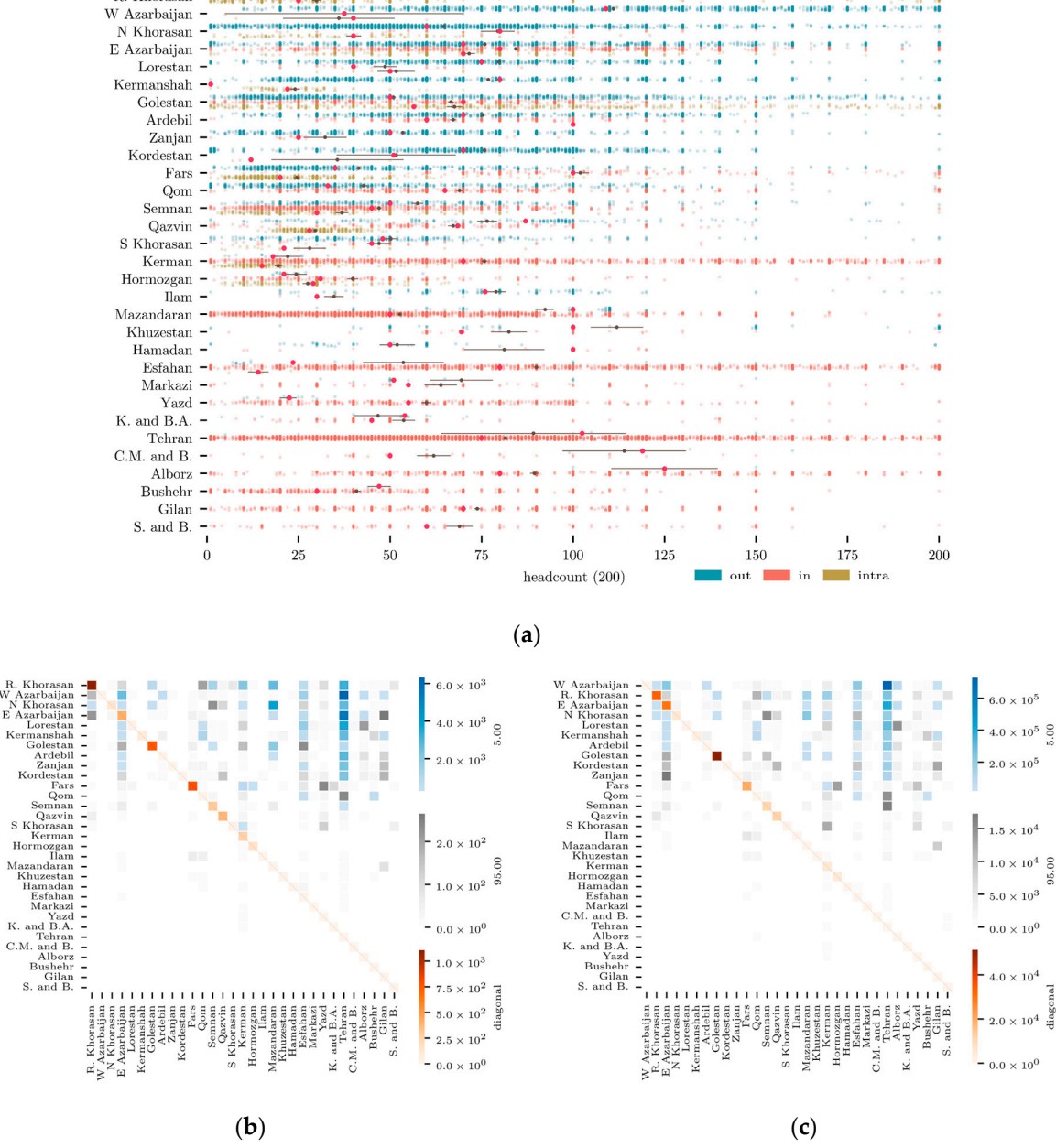

**Figure 8.** Small ruminant shipments in Iran for slaughter only. (**a**) Strip plot of shipments by province and headcount. (**b**) Heatmap of the cumulative number of shipments between pairs of provinces. (**c**) Heatmap of the cumulative volume of small ruminants moved between pairs of provinces.

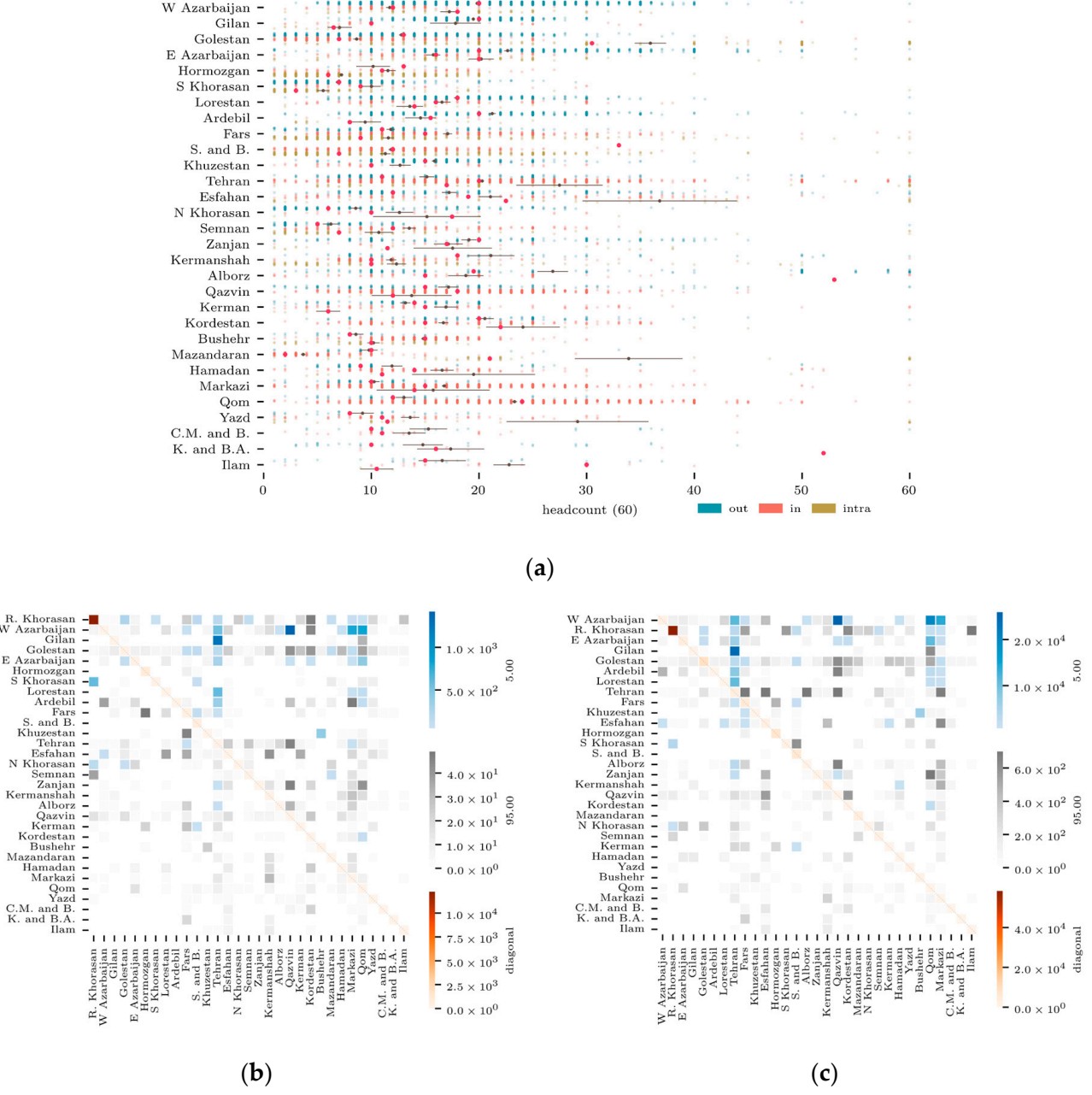

**Figure 9.** Cattle shipments in Iran for farming only. (**a**) Strip plot of shipments by province and headcount. Volumes on the *x*-axis are expressed in heads of cattle, capped at 60 to aid visibility. Red points indicate the median, and the darker-colored mean is accompanied by its corresponding standard error; they correspond to the uncapped data. (**b**) Heatmap of the cumulative number of shipments between pairs of provinces. (**c**) Heatmap of the cumulative volume of cattle moved between pairs of provinces.

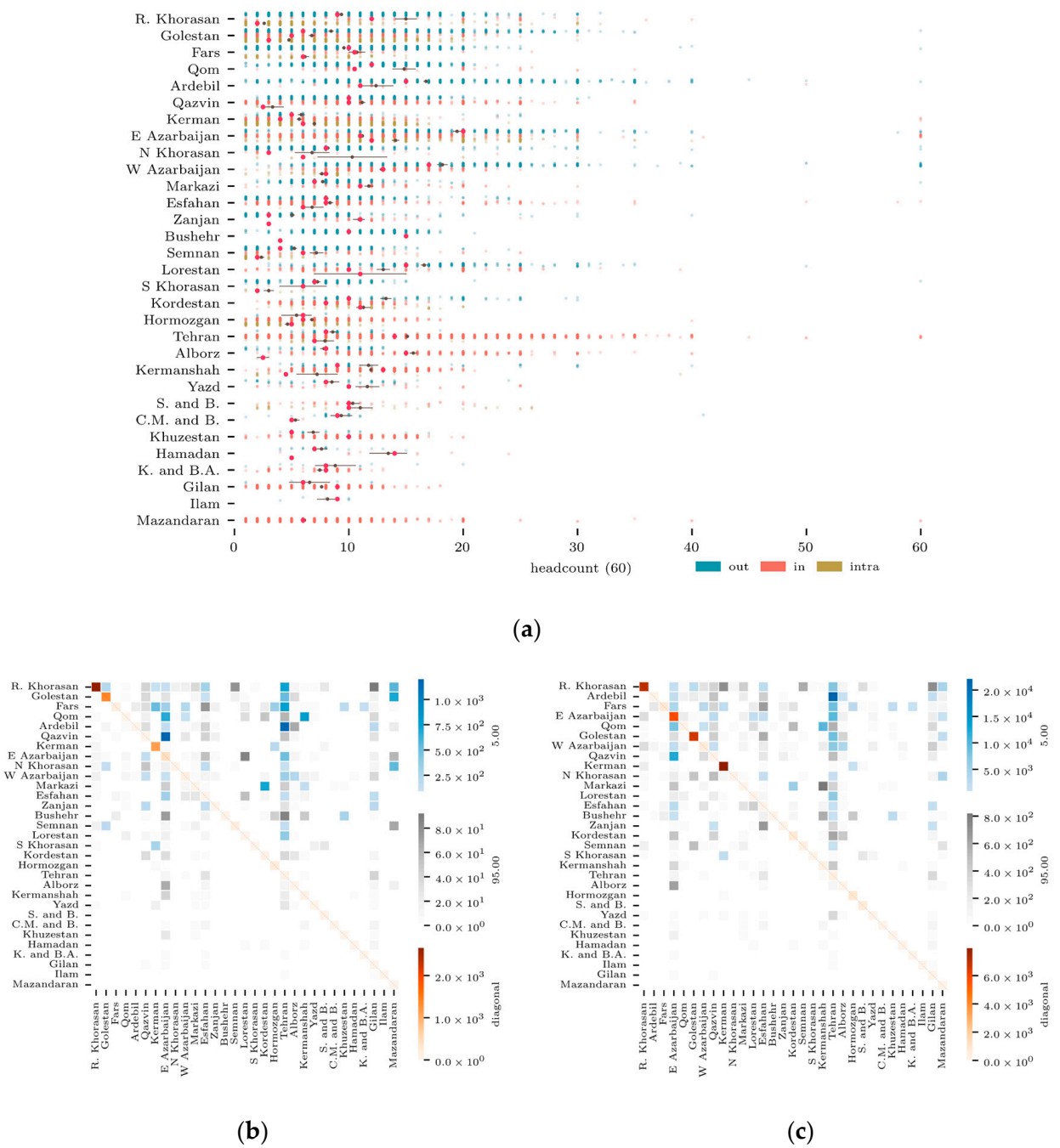

**Figure 10.** Cattle shipments in Iran for slaughter only. (**a**) Strip plot of shipments by province and headcount. (**b**) Heatmap of the cumulative number of shipments between pairs of provinces. (**c**) Heatmap of the cumulative volume of cattle moved between pairs of provinces.

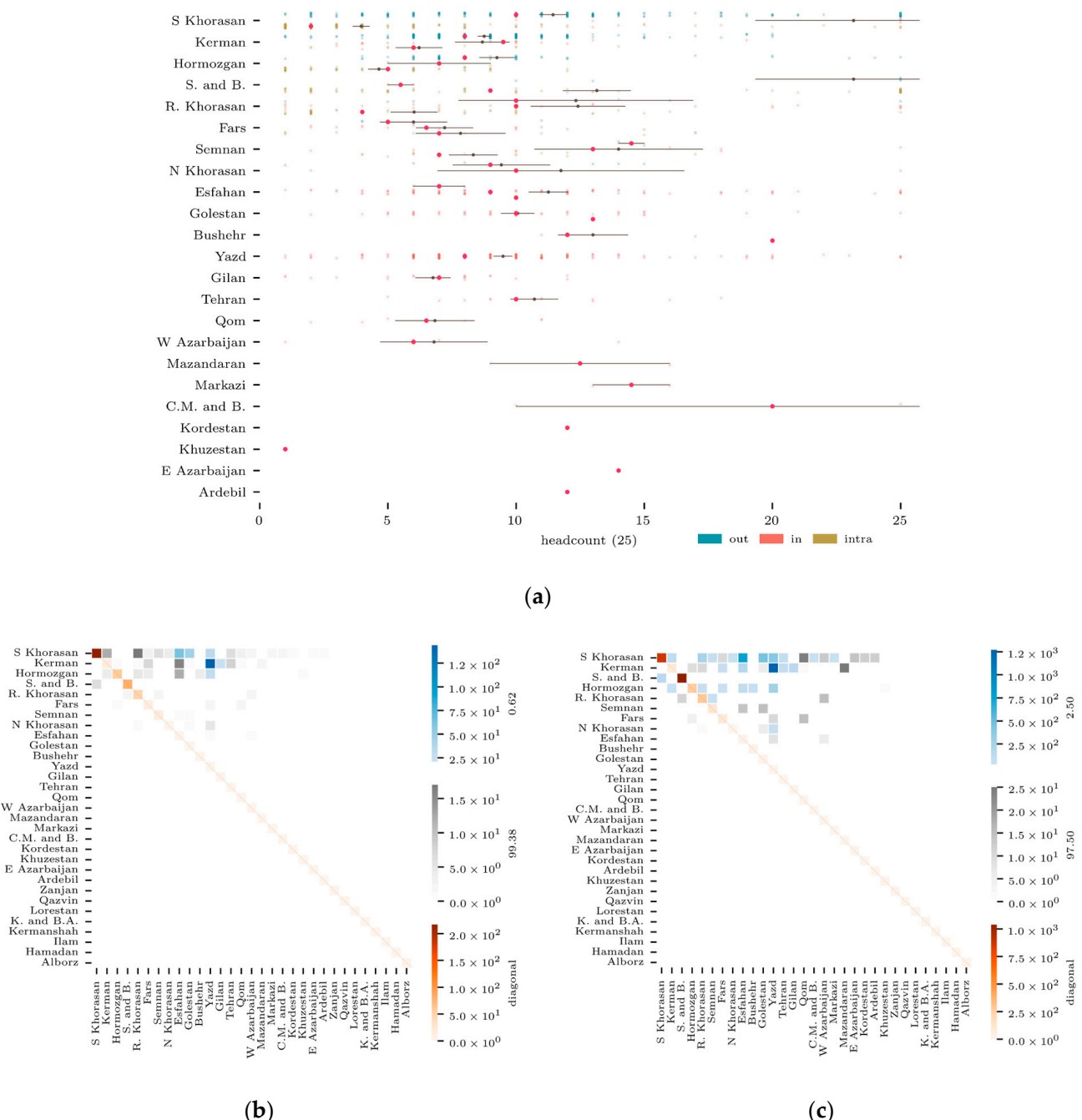

**Figure 11.** Camel shipments in Iran for farming only. (**a**) Strip plot of shipments by province and headcount. Volumes on the *x*-axis are expressed in heads of camels, capped at 25 to aid visibility. Red points indicate the median, and the darker-colored mean is accompanied by its corresponding standard error; they correspond to the uncapped data. (**b**) Heatmap of the cumulative number of shipments between pairs of provinces. (**c**) Heatmap of the cumulative volume of camels moved between pairs of provinces.

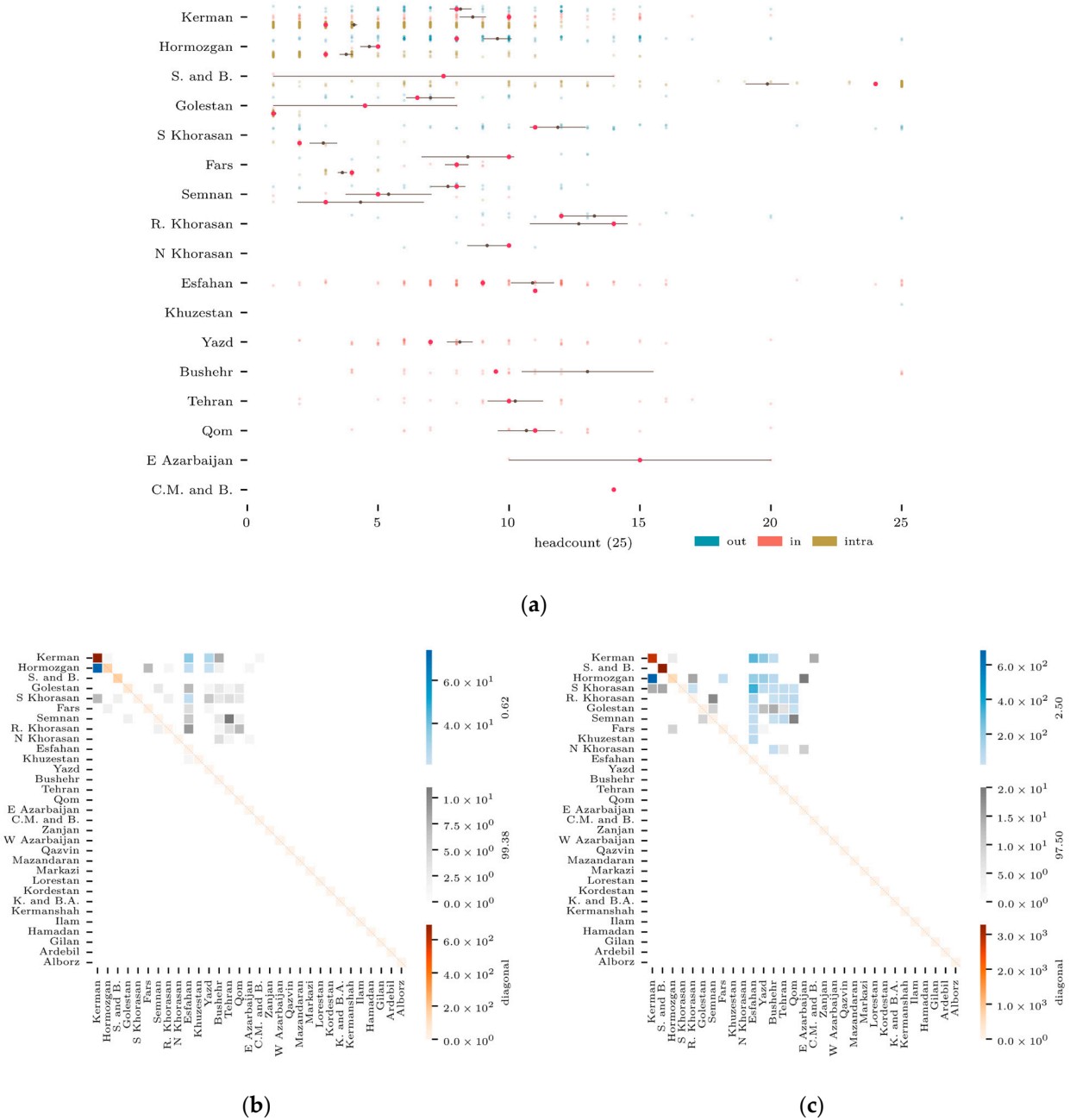

**Figure 12.** Camel shipments in Iran for slaughter only. (**a**) Strip plot of shipments by province and headcount. (**b**) The cumulative number of shipments between pairs of provinces. (**c**) The cumulative volume of camels moved between pairs of provinces.

Razavi Khorasan and West Azerbaijan are the provinces with the highest frequency and volumes of small ruminants moved to slaughterhouses (Figure 8a–c); mean volumes of the small ruminant records for these two provinces range from 25 to 108 (irrespective of in, out, intra categories). The mean volume of small ruminants moved to slaughterhouses varies between provinces (Figure 8a). Tehran province, which hosts the capital city, is one of the most important destinations (movement in only) for small ruminants to slaughterhouses, sourcing small ruminants from 12 different provinces (Figure 8b,c). Razavi Khorasan, Golestan (in Northeastern Iran), and Fars are the three provinces that record the highest numbers of intra-provincial movements of small ruminants to slaughterhouses (Figure 8b,c; Supplementary Figure S2).

Cattle

Razavi Khorasan and West Azerbaijan provinces export cattle mostly to farms (in frequency and volumes), while Tehran, Qom, and Markazi (all three provinces located in central Iran) are the top importing provinces (Figure 9a–c). The mean numbers of cattle moved to farms vary between provinces and movement types and range from 2 to 53 cattle, irrespective of in, out, or intra categories (Figure 9a). Razavi Khorasan, West Azerbaijan, Golestan, or Qom are among the provinces having the three types of cattle movement to farms (Figure 9a). In Razavi Khorasan province, there is intense intra-movement of cattle for farming (Figure 9b,c). The network of cattle movement to farms is less dominated by certain provinces compared to the network of small ruminant movement to farms (Figures 7 and 9; Supplementary Figure S3).

Razavi Khorasan and Golestan are the main exporting provinces in frequency of cattle to slaughterhouses, while Razavi Khorasan and Ardebil (in Northwestern Iran) are the main exporting provinces in volumes (Figure 10a). The mean volume of cattle moved to slaughterhouses varies between provinces but seems lower than the mean volume of cattle moved to farms. The mean volumes range from 2 to 20 cattle, irrespective of in, out, or intra categories. As described for the small ruminants, Tehran and E. Azerbaijan (the latter in Northwestern Iran) are among the main importing provinces for cattle destined for slaughter (Figure 10b,c). Razavi Khorasan, E. Azerbaijan, Golestan, and Kerman show intense intra-provincial movements of cattle for slaughter (Figure 10b,c; Supplementary Figure S3).

Camel

The network for camel movement to farms involves 23 provinces compared to the small ruminant and cattle ones, which both involve 31 provinces, respectively (Figures 7a, 9a and 11a). South Khorasan (in the East of Iran) and Kerman are the top exporting provinces in frequency and volumes of camel moved to farms. The mean volume of camel records to farms varies between provinces, ranging from 1 to 20 camels, irrespective of in, out, or intra categories (Figure 11a). In S. Khorasan and Sistan and Baluchestan provinces (the latter located in Southeast Iran), there are high volumes of camels moved for farming within the province; Yazd province (central Iran) is the top destination for camels moved from Kerman province (Figure 11b,c; Supplementary Figure S4).

Kerman and Hormozgan (South Iran) are the top exporting provinces for camels to slaughterhouses in frequency, while Kerman, Sistan, and Baluchestan are the top exporters in volumes (Figure 12a). The mean volume of camels moved varies between the 15 provinces recording movements to slaughterhouses, with mean values ranging from 1 to 24 camels, irrespective of in, out, or intra categories. In Kerman, Sistan, and Baluchestan, there are also intense intra-provincial movements of camels to slaughterhouses (Figure 12b,c). See supplementary Figure S4 for chord diagrams of the camel movements between provinces.

### 3.2.2. Town Centrality Estimates

We report the hybrid source- and sinkness scores of the top 10 nodes of the farming frequency networks for different species, where the initial scores are augmented by their corresponding authority and hub score through Equation (3). As previously mentioned, towns with higher sourceness scores are estimated to be better candidates for targeted disease prevention investment under constrained resources. Similarly, towns with higher sinkness scores are estimated to be more suitable for targeted detection (sentinel placement).

Table 2 shows the fractions of aggregated edges and shipments ending in exporting (shipping) nodes as well as the fraction of exporting nodes in the static farming frequency networks per species. It can be seen that only a small fraction of the nodes of these networks participate in exporting livestock and, as a result, have nonzero outdegrees. Moreover, only a small fraction of these shipments are made to the exporting nodes; this suggests that the structures of these networks are *nearly* bipartite, ignoring the small fractions of shipments

to exporting nodes. In particular, all the transitivity in these networks is due to (a subset of) these small fractions of edges among the small fractions of their exporting nodes. Moreover, of every directed path of length $n$, the first $n - 1$ edges are confined to among these small fractions of exporting nodes. The following centrality estimates should be seen in this light.

**Table 2.** Fractions of aggregated (weighted) edges and shipments (unaggregated edges) ending in exporting nodes and those of the number of exporting nodes of the static farming frequency networks for different species.

| Species | To-Exporter % Agg. Edges | To-Exporter % Shipments | Exporters % |
|---|---|---|---|
| Small ruminants | 8.11 | 10.77 | 8.12 |
| Cattle | 10.78 | 13.72 | 9.28 |
| Camel | 2.94 | 1.07 | 10.89 |
| All three | 9.51 | 11.74 | 7.54 |

For all species, the highest sourceness score belongs to the town of Orumiyeh (West Azerbaijan province), and the town with the highest sinkness score is Qom (Qom province) in central Iran (Table 3).

**Table 3.** Top ten nodes (towns) in the farming frequency networks considering all three species with the highest source- and sinkness scores. Towns with higher sourceness scores are estimated to be better candidates for disease prevention and control intervention. Towns with higher sinkness scores are estimated to be more suitable for targeted disease detection.

| Source Town | Score | Sink Town | Score |
|---|---|---|---|
| Orumiyeh | $1.00 \times 10^0$ | Qom | $1.00 \times 10^0$ |
| Baneh | $7.06 \times 10^{-1}$ | Khomeini Shahr | $8.33 \times 10^{-1}$ |
| Ahar | $7.00 \times 10^{-1}$ | Ashkezar | $8.32 \times 10^{-1}$ |
| Ardabil | $6.90 \times 10^{-1}$ | Zahedan | $8.07 \times 10^{-1}$ |
| Bandar-e Gaz | $6.64 \times 10^{-1}$ | Rey | $5.81 \times 10^{-1}$ |
| Taybad | $6.63 \times 10^{-1}$ | Mashhad | $5.70 \times 10^{-1}$ |
| Abhar | $6.37 \times 10^{-1}$ | Mehriz | $4.43 \times 10^{-1}$ |
| Aligudarz | $6.06 \times 10^{-1}$ | Fasa | $4.16 \times 10^{-1}$ |
| Tehran | $5.56 \times 10^{-1}$ | Garmsar | $3.83 \times 10^{-1}$ |
| Avaz | $5.52 \times 10^{-1}$ | Varamin | $3.65 \times 10^{-1}$ |

In the small ruminants farming frequency network, the highest sourceness scores are shared between the towns of Baneh (Kurdistan province, in West Iran), Orumiyeh (West Azerbaijan province), and Taybad (Razavi Khorasan province), located near the borders with Iraq, Türkiye, and Afghanistan, respectively. Towns with the highest sinkness score are Khomeini Shahr (Esfahan province), in the central part of the country, and Zahedan (Sistan and Baluchestan province) in Southern Iran (See Supplementary Table S1).

In the cattle framing frequency network, Orumiyeh (West Azerbaijan province) is showing by far the highest sourceness score, while Dashtestan (Bushehr province) in Southern Iran next to the Persian Gulf is showing by far the highest sinkness score (See Supplementary Table S2). It is worth noting that Orumiyeh (West Azerbaijan province) ranked as one of the highest source towns for both cattle and small ruminants farming networks.

Finally, in the camel farming network, similarly to the cattle network, only one town is outscoring. The highest sourceness score belongs to the town of Birjand, while the highest sinkness score is shown by Nehbandan. Both towns are located in South Khorasan province in Eastern Iran, close to the border with Afghanistan, where camel production and trade are concentrated (See Supplementary Table S3).

3.2.3. Lateral Trajectories of Records

Figure 13 depicts the longitudinal and latitudinal differences between destination and origin towns. The recorded movements of camels tend to be directed North–Westwards, while recorded small ruminants and cattle movements are more equally distributed in all directions and of greater distance (Figure 13). Short-distance movements (up to 400 km) of small ruminants are more frequently observed (maximum distance of 1600 km) (Supplementary Figure S5). A similar pattern is observed in cattle and camel mobilities for which short-distance movements (up to 300 km and up to 250 km, respectively) are more frequently observed (maximum distance: 1500 km and 1600 km, respectively) (Supplementary Figure S5).

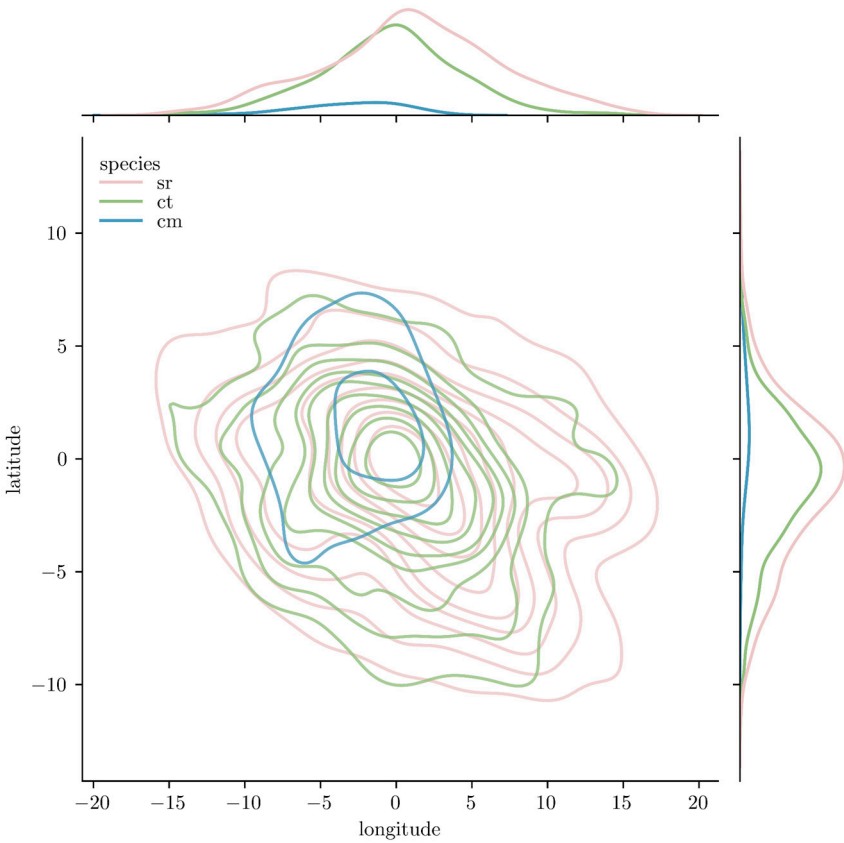

**Figure 13.** Plot of the longitudinal and latitudinal differences between destination and origin towns for small ruminants, cattle, and camel static networks.

## 4. Discussion

Animal mobility is generally hard to describe in low- and middle-income countries because of the limited availability of movement records over an extended period of time. This study is the first spatiotemporal description of nationwide livestock mobility in Iran. It confirms that livestock mobility is an important feature of livestock production and market chains in Iran (both in the volume of records and the number of animals moved), particularly for small ruminants, cattle, and camels. It shows that the static networks drawn by official (registered) movements of livestock are complex, involving the exchange of large volumes of small ruminants and cattle between many provinces across Iran and fewer provinces for the camel network. All networks are dominated by a few provinces. For the farming frequency networks, the top ten *source* and *sink* towns have been identified.

The outputs of our analysis, along with other risk factors for disease transmission and considerations of economic impact are intended to inform the risk assessment of the spread of contagious animal diseases within Iran for which (i) small ruminants, cattle, and camel are susceptible species, and (ii) live animal movement is recognized as a risk factor

for pathogen dissemination [10], such as FMD and similar transboundary animal (FAST) diseases. Such an approach is particularly important in low- and middle-income countries, where animal disease epidemics have strong socioeconomic impacts [13] and the resources for disease control are limited. Trade barriers and movement restrictions at the local level, despite substantial economic consequences for the affected stakeholders, are some of the main components of FMD control [21,22].

Our analysis indicates that the Razavi Khorasan province, a major one for the Iranian livestock sector, frequently exports significant volumes of small ruminants and cattle to and from the rest of Iran and is also the setting for intense intra-provincial movements. It, therefore, may be prioritized for targeted and timely risk reduction interventions, such as a temporal increase in movement controls in case of uncontrolled sanitary situations, an increase in biosecurity levels at farms, at markets, or during transportation, vaccination prior to any movement. Risk reduction interventions could also be suggested in West Azerbaijan province, including the town of Orumiyeh, which shows the highest *source score* in the farming network considering cattle, sheep, goat, and camel mobilities. Active surveillance of FAST diseases is a cost-efficient approach for monitoring circulating strains in endemic settings. Such information is crucial for regional preparedness and vaccine selection. A prophylactic six-monthly vaccination of large and small ruminants against FMD is currently implemented in Iran, with vaccines including serotypes O, A, and Asia1. Given the situation with the exotic FMDV SAT2 in neighboring countries, surveillance for SAT2 early detection in Iran is paramount and would contribute to reducing the impact of such an incursion into naive livestock populations.

Our analysis of the farming networks points out a subset of towns widely distributed across Iran and shows the highest *sink scores*. These towns, and specifically in premises where moved animals are gathered, could be suitable areas for active surveillance of FAST pathogens. Considering the complexity and magnitude of small ruminant mobility in Iran, that FMD could spread silently through small ruminant movements because of mild clinical signs, and recent study outcomes identifying small ruminant populations as a significant factor influencing disease prevalence (refs to be inserted), active surveillance targeting small ruminants could be advised. It is, however, important to stress that the data available did not include informal (non-registered) and illegal animal movements nor international (including cross-border) movements of livestock. Cross-sectional studies targeting actors involved in national and cross-border informal or illegal livestock mobility could be envisaged to complement the current description and reassess the top *sink* towns accordingly.

Our descriptive analysis indicated that small ruminant mobility to farms and slaughterhouses peaked (in volume and frequency) right before Tabaski celebrations, and the network also gained connectedness on these occasions. It is speculated that some of the small ruminants moved to farms right before Tabaski were destined for sacrifice out of a slaughterhouse. Moreover, in many provinces of Iran, lambing mostly occurs during the last month of every Jalali year (the month of Esfand corresponds to the end of February up to the end of March). Small ruminant owners usually maintain the lambs for about six months and sell them as fattened lambs from July to September, as at that time, pastures gradually become exhausted. Iranian nomads move their livestock (mostly small ruminants) to grazing lands from May to September.

A decline in movement records was observed for small ruminants and cattle during Nowruz, the Persian New Year. Nowruz signals the emergence of green pastures, and owners utilize them to reduce feeding costs, leading to a stark reduction in shipments around the time of the occasion. This may impact the functioning of the value chain and imply a reduction in the volume of animals moved around mid-March 2021. However, as our data only cover a one-year period, a trend cannot be conjectured.

Cultural, socioeconomic, and environmental drivers of livestock mobility are plural (including retail price fluctuations, climatic conditions, forage availability, the occurrence of disease outbreaks, or may depend on national and global economic situations). This

study can be a starting point for further research, such as spatiotemporal modeling of current movement patterns and the effects of measures such as targeted vaccination in Iran. A finer characterization of types of origins and destinations, in particular, for the type of farms (traditional, semi-industrial, industrial), livestock markets, and slaughterhouses, would allow for a greater reconstruction of livestock production and trading chain in Iran by distinguishing two segments as a minimum, the one related to smallholder type of production and trade, and the one related to the semi-industrial/industrial type. It is speculated that each segment would be characterized by specific actors and premises, biosecurity practices, and livestock mobility ruled by specific drivers within this segment. Livestock dealers represent a key group of trading actors in Iran, buying and selling livestock in the villages and transporting them either locally or distantly [7,11]. Depending on the price of live animals, dealers can transport a large number of livestock between different provinces, especially between central Iran (Qom province, in particular) and the Western, Northeastern, and Southeastern provinces. The available data could not allow for the identification of the actor(s) behind the movements recorded (either pastoralists, sedentary farmers, or dealers). This might have helped to assess their respective importance within the production and marketing chains, as well as to target communication and awareness for disease reduction strategies. In addition, our analysis showed that 35.5% of small ruminant movement records originate from livestock markets, corresponding to 32.4% of the total number of small ruminants moved (Table 1). This paper does not describe livestock mobility per origin type but rather per destination type; it, however, acknowledges the usefulness of the specific study patterns of livestock mobility from livestock markets in Iran. Finally, the data available did not allow for the study of multi-stop itineraries of animals, which could be of particular relevance for livestock movements "for farming". This would have required a traceability system at the batch or animal level, which is currently not being recorded in the quarantine system in Iran.

Although it is observed that records of livestock mobility tend to increase during the study period, a trend that is confirmed by the published statistics of the Ministry of Agriculture of Iran (https://www.amar.org.ir (accessed on 20 September 2023)) indicating an increase in livestock slaughter from 2019 to 2021, this study was not intended to assess the impact of the release of COVID-19 restrictions on livestock mobility. The extent to which the Iranian livestock value chains have changed the way they operate (including livestock mobility along these chains) during and following the COVID-19 pandemic remains to be studied for updating national disease control strategies.

## 5. Conclusions

This study provides baseline information to aid the risk assessment of the spread of contagious animal diseases in Iran, such as FMD. Regular risk assessments will assist national decision-making through the identification of hotspots and the subsequent development of risk-based surveillance strategies and early warning systems, allowing for the best use of limited resources and improving the effectiveness of the implemented control measures. Risk assessment outputs are also useful for disease control and prevention advocacy.

**Supplementary Materials:** The following supporting information can be downloaded at: https://www.mdpi.com/article/10.3390/ruminants3040027/s1. Figure S1. Chord diagram of the small ruminant networks; Figure S2. Chord diagram of the cattle networks; Figure S3. Chord diagram of the camel networks; Figure S4. Scatter plots of the longitudinal and latitudinal differences between destination and origin towns; Figure S5. Maps showing the distribution of farms, livestock markets and slaughterhouses (aggregated at town level) in Iran; Table S1: Top 10 nodes (towns) in the small ruminant farming frequency networks with highest source- and sickness scores; Table S2: Top 10 nodes (towns) in the cattle farming frequency networks with highest source- and sickness scores; Table S3: Top 10 nodes (towns) in the camel farming frequency networks with highest source- and sickness scores.

**Author Contributions:** Conceptualization, K.M.; data curation, S.M.; formal analysis, S.M.; methodology, S.M.; software, S.M.; supervision, F.A. and F.R.; validation, E.A., B.V.A. and E.C.; visualization, S.M.; writing—original draft, K.M., S.M., E.A. and E.C.; writing—review and editing, E.A., B.V.A. and F.A. All authors have read and agreed to the published version of the manuscript.

**Funding:** This research received no external funding.

**Informed Consent Statement:** Not applicable.

**Data Availability Statement:** Restrictions apply to the availability of these data. Data were obtained from the Iran Veterinary Organization (IVO) and are available by request directly from the quarantine bureau of IVO.

**Acknowledgments:** This work was initiated in the "Risk mapping for improved FMD and similar transboundary animal diseases surveillance and early detection" regional training course, supported by the European Commission for the control of foot-and-mouth disease (EuFMD) in the framework of the Risk reduction program (Phase V 2019–2023) and facilitated by the International Center for Agronomic Research and Development (CIRAD), research unit ASTRE. Mirzaie is grateful to Seyed Mohammad Aghamiri and Vahid Salehi, the previous and current Chief Veterinary Officers of the Islamic Republic of Iran and Seyed Hedayat Hosseini, Senior Advisor of the IVO, for official authorization to conduct scientific works in this context. S.M. would like to thank Karim Jami-al-Ahmadi for imparting valuable insight into the seasonal trends in the region.

**Conflicts of Interest:** The authors declare no conflict of interest.

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
