# Peer review of "Analysis of Livestock Mobility and Implications for the Risk of Foot-and-Mouth Disease Virus Spread in Iran"

_ruminants, doi:10.3390/ruminants3040027_

Round 1

Reviewer 1 Report

I enjoyed reading the manuscript entitled as "Network analysis of livestock mobility and implications for the  risk of foot–and-mouth disease spread in Iran" authored by Kamran Mirzaie  et al.,. This manuscript is a nice one and gives a very good understanding for the readers interested in the area, the policy makers in Iran and the scientific community. I recommend this article to be published in your journal considering its importance and originality. I have one comment to the author that the impact of illegal animal movements have never been mentioned in the manuscript, if possible try to estimate that or state as a limitation of this study.

Author Response

Response: Dear Reviewer, we do appreciate your positive feedback and suggestions for discussion.

It was already stressed in the discussion that “the data available did not include informal (non-registered) animal movements, nor international (including cross-border) movements of livestock. Cross-sectional studies targeting actors involved in informal and cross-border livestock mobility could be envisaged to complement the current description, and re-assess the top sink towns accordingly”.

In our understanding, non-registered and informal movements include illegal ones. However, we took good note of your suggestion and amended the study limitations (within the discussion section) accordingly, by adding the term“illegal”. Thank you.

Reviewer 2 Report

1.       Introduction:

The topic of the manuscript by Mirzae et al. “Network analysis of livestock mobility and implications for the risk of foot-and-mouth disease spread in Iran” is highly relevant as it  provides a spatio-temporal description of susceptible foot-and-mouth disease (FMD) livestock mobility networks in Iran. The study highlights the significance of livestock mobility as a key risk factor for FMD virus transmission in an endemic setting.

2.       Material and Methods:

This section needs significant improvement to enhance clarity. It should focus on explaining the choices made and reasons behind them in a simpler manner. This includes the aggregation model, network parameters, metrics, and software used. Additionally, the meaning of each element in the equations should be explained clearly, as well as the meaning of all terms. For example: “nodes”, “edges”, “referral” in line 178, “lower topological resistance against nodes transitively influencing each other”, “fixing the first/second index” in lines 197/200, “hub and authority centralities”, “middle-shifted degree-weighted versions of the notions of source and sink”, “HITS algorithm”,  or “semantic content of this attribute” in line 207). Also, the reasons for reversing orientation of edges, fixing indexes, reweighting of efficiency scores, ranking or newly ranking nodes, should be provided. See the following article as an example, where network analysis terminology is explained in a more reader-friendly manner: https://www.ncbi.nlm.nih.gov/pmc/articles/PMC7201065/.

It is not clear which elements are included in the network, such as farm-farm, village-farm, village-slaughter, farm-slaughter connections, and the absence of markets or traders need clarification. Additionally, the total number of animals per species detailed at the corresponding epidemiological unit should be considered for better interpretation.

3.       Results:

The presentation of mean results is useful, but additional information on the distribution (quartiles, median, min, max) would enhance understanding. Graphics should be interpreted considering the underlying population, stratifying by province or focusing on provinces with higher number of movements/livestock volume or farms. The distribution of slaughterhouses and farms across Iran should also be included.  

Some paragraphs in the Results section could be moved to the Discussion section or included in the Supplementary Material file to make the manuscript more reader-friendly.

4.       Figures:

Figure 1: clarify the meaning of the number at the top and in the middle of each bar. Remove “other” and “buffalo” from the figures since they were not kept for further analysis. Indicate the proportion of movements to slaughterhouses as stated in line 120 and in the caption of the figure. The figure could be placed under the results section.

Figure 4 should have the meaning of the legends and dashed/solid lines in the caption rather than in the text. Figures 4 and 5 can be redesigned to separate records to slaughterhouses and records to farms into different graphics, including the normalized curve in each. Normalization should also consider variations in frequency, not just volume.

5.       Minor comments: Line 34: Genus in italics; use FMD /FMDV consistently (revise title, lines 43, 104, etc.); correct in line 94: Organisation with s for WOAH.

Rather than the quality of English language, the difficulty in reading this manuscript reveals the need for improved clarity and organization. The manuscript lacks sufficient explanations and context for certain terms and concepts, making it hard for readers unfamiliar with network analysis to understand the study and its findings. Additionally, the Material and Methods section requires significant improvement in explaining the methodology in a straightforward manner, focusing on the choices made and their justifications.

Author Response

Reviewer 2:

Introduction: The topic of the manuscript by Mirzaie et al. “Network analysis of livestock mobility and implications for the risk of foot-and-mouth disease spread in Iran” is highly relevant as it provides a spatio-temporal description of susceptible FMD livestock mobility networks in Iran. The study highlights the significance of livestock mobility as a key risk factor for FMD virus transmission in an endemic setting.

Material and methods: The section needs significant improvements to enhance clarity. It should focus on explaining the choices made and reasons behind in a simpler manner. This includes the aggregation model, network parameters, metrics, and software used. Additionally, the meaning of each element in the equations should be explained clearly, as well as the meaning of all terms. For example: ‘nodes”, ‘edges’, ‘referral’ in line 178, ‘lower topological resistance against nodes transitively influencing each other’, ‘fixing the first/second index’ in lines 197/200, ‘hub and authority centralities ‘, ‘middle-shifter degree-weightened version of the notions of sources and sinks’, ‘HITS algorithm”, or ‘semantic content of this attribute’ in line 207. Also, the reasons for reversing orientation of edges, fixing indexes, reweighting of efficiency scores, ranking or newly ranking nodes, should be provided. See the following article as an example where network analysis is explained in a more reader-friendly manner https://www.ncbi.nlm.nih.gov/pmc/articles/PMC7201065/

Response: Dear Reviewer, we tried to make our point clearer in the Material and methods (and here below), and we improve for clarity the entire section.

  • aggregation models - aggregation is relevant so that model construction can become faster and more intuitive while holding true to the apparent structure of the Iranian animal mobility being modeled.
  • nodes (origin and destination points of the network, either towns or provinces)
  • edges, i.e., links between two nodes (origin and destination points of the network)
  • We used the hybrid source and sink scores, to estimate the relative importance of certain towns in the network of animal mobility in Iran. In the context of infectious disease spread, towns with high (hybrid) sourceness centrality have a high amount of outgoing movements to various destinations and can indicate locations where the virus can more easily and more quickly spread from. Towns with high (hybrid) sinkness centrality have a relatively high amount of incoming traffic from various origins, making them particularly susceptible to pathogen introduction.
  • In the context of infectious disease spread, the hybrid centralities allow us to identify influential nodes in the animal mobility network.
  • Software: all analyses were done using the Python programming language

It is not clear which elements are included in the network, such as farm-farm, village-farm, village-slaughter, farm-slaughter connections, and the absence of market or traders need clarifications. Additionally, the total number of animals per species detailed at the corresponding epidemiological unit should be considered for better interpretation.

Response: Dear Reviewer, we considered livestock movements from either farm or livestock market to farms or slaughterhouses. The analysis was stratified by species, and aggregated either at the provincial or town levels. The mobility to farm or to slaughterhouse was described, however, we have not considered, due to resource constraints, the study of the mobility based on the origin type (from farms or markets). Table 1 has been enriched with metrics on edges and volumes based on the origin type. Also, the discussion section has been edited to reflect on the current limitations of our study (wrt the study of livestock mobility from livestock markets).

Results: The presentation of mean results is useful, but additional information on the distribution (quartiles, median, min, max) would enhance understanding. Graphics should be interpreted considering the underlying population, stratifying by province or focusing on provinces with higher number of movements/livestock volumes or farms. The distribution of slaughterhouses and farms across Iran should also be included.

Response: Dear Reviewer, thank you. We have added maps showing the distribution of farms, livestock markets, and slaughterhouses (aggregated at the town level) across Iran as supplementary material.

Some paragraph in the Results section could be moved to the Discussion section or included in the Supplementary Material file to make the manuscript more reader-friendly.

Response: Dear Reviewer, we have revised the entire manuscript, and hopefully it is more reader-friendly now.

Figure 1: clarify the meaning of number at the top and in the middle of each bar. Remove ‘other‘ and ‘buffalo’ from the figure since they were not kept for further analysis. Indicate the proportion of movements to slaughterhouses as stated in line 120 in the caption of the figure. The figure could be placed in the results section.

Response: Dear Reviewer, we have removed “buffalo” and “others” from the figure 1 (now figure 2 in the revised manuscript, as this figure has been placed in result section as suggested). We thank you for this suggestion that makes the manuscript clearer. We have edited the caption to ease the reader’s understanding of the figure, with particular reference to the number accompanying bars.

Figure 4: should have the meaning of the legends and dashed lines in the caption rather than in the text. Figure 4 and 5 can be redesigned to separate records to slaughterhouses and records to farms into different graphics. Normalization should also consider variation in frequency, not just volume.

Response: Dear Reviewer, we thank you for your suggestion. We had, in the past, designed figures 4 and 5 separating records to farms and to slaughterhouses. However, we have realized that separating such records makes the visualization of ‘parallel trends’ more difficult. Hence, we decided to keep the two types of records in the same figures.  In addition, please be informed that explanations for dashed and solid lines are presented in the caption of Figure 4, 5 and 6.

Line 34: genus in italics – addressed thanks; use FMD/FMDV consistently (revise title  - addressed thanks, lines 43, 104 etc); correct in line 94: Organisation with s for WOAH - addressed thanks.

Rather than the quality of English language the difficulty in reading this manuscript reveals the need for improved clarity and organization. The manuscript lacks sufficient explanations and context for certain terms and concepts, making it hard for readers unfamiliar with network analysis to understand the study and its findings. Additionally, the Material and Method sections requires significant improvement in explaining the methodology in a straightforward manner, focusing of the choices made and their justifications. 

Response: The manuscript has been English edited.

Reviewer 3 Report

This is a very comprehensive paper.  The findings are valuable for national policy of Iran by identifying main linkages between different regions in animal movements.  To be even more useful it requires further research, such as spatio-temporal modelling of current movement patterns, and effects of measures such as concentrating vaccination at particular locations.  But this is an excellent starting point.

One issue of concern to me is the use of the basic reproduction number in relation to animal movement rather than disease transmission, when the focus of the paper is on foot and mouth disease risk, but there is no disease data reported.  I am aware that this measure is sometimes used beyond infectious disease transmission (Sisk and Fefferman 2022), but I find it confusing to have it used in relation to animal movement alone in this paper.  I could not find a clear interpretation of the meaning of the term and Figure 7 in the context of this paper in either results or discussion. The authors should either provide a much better explanation of how they have used the term and how the reader should interpret Figure 7, or reconsider its use since it has such a well recognized meaning in disease transmission.

The paper is generally very well written, there are some phrases and sentences which require reading 2 o 3 times and could be made easier to read, but these are quite minor.

Author Response

Reviewer 3: This is a comprehensive paper. The findings are valuable for national policy in Iran by identifying the main linkages between different regions in animal movements. To be even more useful it requires further research, such as spatiotemporal modeling of current movement patterns, and effects of measures such as concentrating vaccination at particular locations. But this is an excellent starting point.

Response: Dear Reviewer, we want to thank you for your positive review and valuable insights. This manuscript is indeed the starting point for further research on FMDV dynamics in Iran.  This article is meant to lay the groundwork for risk assessment of FMD and similar transboundary animal diseases’ spread within Iran, to inform decision-making in disease control (at central and provincial levels), the latter including vaccination. Modeling could also be envisaged, in a later stage.

We have also added a phrase on your idea as future work: “To be even more useful it requires further research, such as spatiotemporal modeling of current movement patterns, and effects of measures such as concentrating vaccination at particular locations”. Thank you.

One issue of concern, to me is the use of the basic reproduction number in relation to animal movement rather than disease transmission, when the focus of the paper is on FMD risk, but there is no disease data reported. I am aware that this measure is sometimes used beyond infectious disease transmission (Sisk and Fefferman, 2022), but I find it confusing to have it used in relation to animal movement alone in this paper. I could not find a clear interpretation of the meaning of the term and Figure 7 in the context of this paper in either results or discussion. The authors should either provide a much clearer explanation of how they have used the term and how the reader should interpret Figure 7 or reconsider its use since it has such a well-recognized meaning in disease transmission.

Response: Dear Reviewer, the authors have carefully considered your comment. Although use of the basic reproduction number is relevant in this context from a mathematical standpoint, it may bring confusion for a reader who may have an epidemiological or disease policy background, and indeed since we do not report any disease data. We are in agreement to delete the parts that relate to the reproduction number in methods and results, and consequently the discussion, in the context of this article.

The paper is generally very well written, there are some phrases and sentences which require reading 2 or 3 times and could be made easier to read, but there are quite minor.

Response: Thank you for the observation. A professional English proofreader has gone through he manuscript and has hopefully improved the writing, hope the article has gained in clarity.

Reviewer 4 Report

Dear Authors,

The manuscript entitled "Network analysis of livestock mobility and implications for the risk of foot–and-mouth disease spread in Iran", is an interesting study of the spatiotemporal impact of the mobility of livestock on dissemination of FMD. Please find my comments as follows:

Line 39: I suggest adding an explanation to the introduction about other ruminants diseases that may occur due to livestock transportation and mobility such as bovine tuberculosis (bTB), brucellosis, bovine viral diarrhea (BVD), scrapie, foot-and-mouth disease (FMD) and Johne's disease. Cite the following publications : 

Prentice JC, Marion G, Hutchings MR, McNeilly TN, Matthews L. Complex responses to movement-based disease control: when livestock trading helps. J R Soc Interface. 2017 Jan;14(126):20160531. doi: 10.1098/rsif.2016.0531.

Line 50: You may change this part by adding other examples as follows: " FMD is an endemic condition. According to a study, there should be an association between a higher incidence of FMD and season changes in which the incidence was maximized between January and March. A similar seasonal effect has also been reported for other ruminant diseases such as Paratuberculosis. A study suggested that the higher incidence of sheep infected with Mycobacterium avium subsp. paratuberculosis (MAP), in Spring, shedding the microorganism in milk might be associated with seasonal breeding, possible sexual transmission of MAP, and the presence of other animals shedding MAP via feces.

Cite the added part with the following reference: Hosseiniporgham, S.; Cubeddu, T.; Rocca, S.; Sechi, L.A. Identification of Mycobacterium avium subsp. paratuberculosis (MAP) in Sheep Milk, a Zoonotic Problem. Microorganisms 2020, 8, 1264. https://doi.org/10.3390/microorganisms8091264

Line 43-44: Do you mean the original pool? "virus can spread from their historical pool to another one, with the potential to cause significant outbreaks in livestock not previously exposed to the virus serotype"

The English style is fine. 

Author Response

Reviewer 4: Introduction and cited references ca be improved

Response: Thank you for the suggestion. Please check the revised introduction and cited references.

Dear authors, the manuscript entitled “Network analysis of livestock mobility and implications for the risk of foot-and-mouth disease spread in Iran”, is an interesting study of the spatiotemporal impact of the mobility of livestock on dissemination of FMD. Please find my comments as follows:

Line 39: I suggest adding an explanation to the introduction about other ruminant diseases that may occur due to the livestock transportation and mobility such as bovine tuberculosis, brucellosis, bovine viral diarrhea, crappie, FMD and Johne’s disease. Cite the following publication: https://pubmed.ncbi.nlm.nih.gov/28077759/

Response: Dear Reviewer, this is an excellent suggestion and very relevant publication. We have stressed this idea in the introduction, and discussion, and have cited the suggested publication.  

Line 50: you may change this part by adding other examples as follows: “FMD is an endemic condition. According to a study, there should be an association between higher incidence of FMD and season changes in which the incidence was maximized between January and March. A similar seasonal effect had also been reported for other ruminant diseases such as paratuberculosis. A study suggested the that the higher incidence if sheep infected with Mycobacterium avium subsp. Paratuberculosis (MAP), in Spring, shedding the microorganism in milk might be associated with seasonal breeding, possible sexual transmission of MAP and the presence of other animals shedding MAP via feces”. Cite the following reference: https://www.ncbi.nlm.nih.gov/pmc/articles/PMC7565042/

Response: Dear Reviewer, again thank you very much. A recent paper published in Scientific reports entitled ‘Socioeconomic and environmental determinants of foot and mouth disease incidence: an ecological, cross-sectional study across Iran using spatial modeling’  https://www.nature.com/articles/s41598-023-40865-4 looks very relevant to our study, which findings help us to address your suggestion to elaborate more on determinants influencing observed seasonal patterns of FMD. Thank you.

Line 43-44: do you mean original pool? “virus can spread from their historical pool to another one, with the potential to cause significant outbreaks in livestock not previously exposed to the virus serotype”.

Response: Yes exactly, thanks for the correction - addressed in the document.

Round 2

Reviewer 2 Report

I appreciate the authors’ efforts in addressing my comments but I am afraid I still find the M&M section not sufficiently clear to be reproduced- or maybe I get lost with so much theory around what has actually been done. The manuscript will greatly improve if the sections in the M&M are matched with those presented in Results. Shortening the manuscript would also improve readibility- some sections particularly in M&M are theoretical and could be removed.

INTRODUCTION

Please indicate where each of the provinces mentioned is located and name them in the maps, so the reader unfamiliar with Iran’s provinces can follow the text and figures.

M&M

-          Section 2.1.Dataset: A new sentence has been added indicating that buffaloes and others are excluded from “All species”. The sentence explaining that the label “all” in Figure 2 in the results section refers to small ruminants, cattle and camels should be placed under Figure 2, and not here. In addition, Figure 2 does not seem to reflect this change

-          Section 2.2. Network construction.

o   2.2.1. Aggregation models. The reader does not need a textbook on aggregation models, but a simple explanation that you have chosen an aggregation model that considers frequency of movements and headcounts as weights. If the unital model is trivial, then you can skip it. Keep only the relevant information that make the methods reproducible and justified. How are the pairs ordered? In this section it seems pairs are only made of town-town. This should be made clear, i.e. a sum of the number of farms and slaughterhouses was obtained by town and the network constructed between towns of origin and destination.

o   2.2.2. Dissimilarity inference. There is a lot of theory here and it is not clear what was applied, why and how it is interpreted or applied to your research. Why do you need to infer dissimilarity, with what purpose? Such explanations are missing

o   2.2.3. Orientation reversal. The same comments to 2.2.2 apply here regarding why was it applied, what is being done, why and how it is interpreted.

-          Section 2.3. Static measures.

o   2.3.1. Global clustering coefficient. Please explain what the first two sentences mean and rewrite more clearly. What is Tr in formula (1)? How is this coefficient interpreted? Why are you applying it, with what purpose?

o   2.3.2. Global efficiency- the only bit missing here is why is this measure important

o   2.3.3. Local efficiencies. Why is this restriction needed in addition to global efficiency? Did you fix the first index, the second index or both? How do you fix the indeces? What is the added value of measuring the forward and backward efficiencies?

-          Section 2.4. Hub and authority scores. Here the new text incorporates many of the suggestion I have raised for previous sections. It is not clear however why you need to reweight your modified forward and efficiency scores. In the authors’ response, this has been named hybrid and sink scores. Please clarify.

-          Section 2.5. Hybrid centralities. Can you please summarize which centrality measures are you considering and why. How do you rank the nodes to obtain a and b.

-          Section 2.6. Prevention versus detection. Please clarify what a sink is referring to in the text. Please also clarify and summarize (i.e with a table) which measures have you chosen for your study, why and how they are interpreted of those you mention in a theoretical way: betweenness, weighted outdegree (clarify what are the out-edges of the node and its unweighted outdegree), shortest path, inverse distance, sourceness.

-          Section 2.7. Temporal analysis. It is not clear why of the three common approaches the first was chosen and why, and how it is interpreted. While it is of great information to know that there are three common approaches, the text could be narrowed and simplified to explain better the method chosen, rather than introducing every method available.

-          Section 2.8. Spatial analysis. Which aggregation method was used?

RESULTS.

-          Figure 2. “All” here includes also buffalo and other species, as stated in the legend, or without them, as stated in M&M? It is also still not clear the interpretation of the graph. In the example in the new legend, 29,173 are directed to farms, but the number appears in the light green part of the column, which corresponds to movements to slaughterhouses. Please clarify/correct as necessary.

-          The sentence in lines 346-348 is needed in M&M, not here.

-          Please summarize the towns which have the greatest volume/frequency in Figure 3 in the text, and explain where they are located (province)

-          Please rearrange either M&M sections or Results so that sections under both epigraphs match

-          Section 3.1. Temporal description. Please remind the reader of the timespan of the dataset. None of this has been presented in methods- please include in methods that you are going to describe the temporal evolution in regards of Tabaski and Nowruz, for example. Please rearrange M&M sections to match those of the Results.

-          Section 3.2. Spatial description. You must state in the M&M sections that you will summarize your results as strip plots, that you will stratify your results by species or that you will compare importing/exporting towns-none of this has been stated in M&M

o   Town centrality estimates. Only hybrid source and sinkness  scores are reported- again, an indication that the M&M needs further rearrangements to reflect the results

o   Lateral trajectories of records- where is this in M&M?

The English is fine.

Author Response

Manuscript title: “Network analysis of livestock mobility and implications for the risk of foot-and-mouth disease spread in Iran”

Manuscript number: ruminants-2532800

07/09/2023

Response to the reviewers

We would like to thank the four reviewers for their careful reading and relevant comments on our work. We have tried to address their comments, and suggestions as much as possible, and have provided explanations when needed in this document.

Reviewer 2:

Open Review

(x) I would not like to sign my review report

( ) I would like to sign my review report

Quality of English Language

( ) I am not qualified to assess the quality of English in this paper

( ) English very difficult to understand/incomprehensible

( ) Extensive editing of English language required

( ) Moderate editing of English language required

(x) Minor editing of English language required

( ) English language fine. No issues detected

            Yes      Can be improved       Must be improved     Not applicable

Does the introduction provide sufficient background and include all relevant references?

            (x)        ( )         ( )         ( )

Are all the cited references relevant to the research?

            (x)        ( )         ( )         ( )

Is the research design appropriate?

            (x)        ( )         ( )         ( )

Are the methods adequately described?

            ( )         (x)        ( )         ( )

Are the results clearly presented?

            ( )         (x)        ( )         ( )

Are the conclusions supported by the results?

            (x)        ( )         ( )         ( )

Comments and Suggestions for Authors

I appreciate the authors’ efforts in addressing my comments but I am afraid I still find the M&M section not sufficiently clear to be reproduced- or maybe I get lost with so much theory around what has actually been done. The manuscript will greatly improve if the sections in the M&M are matched with those presented in Results. Shortening the manuscript would also improve readibility- some sections particularly in M&M are theoretical and could be removed.

Response: Dear Reviewer, we sincerely appreciate your specific comments and suggestions that - we believe - helped us to improve the manuscript.

INTRODUCTION

Please indicate where each of the provinces mentioned is located and name them in the maps, so the reader unfamiliar with Iran’s provinces can follow the text and figures.

Response: In our view, the addition of maps will only clutter the article with information that is readily accessible in the public domain. However, we do, now, provide geographic markers where such information becomes relevant for the first time. In addition, maps are provided as supplementary material, indicating names of provinces.  

M&M

- Section 2.1.Dataset: A new sentence has been added indicating that buffaloes and others are excluded from “All species”. The sentence explaining that the label “all” in Figure 2 in the results section refers to small ruminants, cattle and camels should be placed under Figure 2, and not here. In addition, Figure 2 does not seem to reflect this change

Response: We have reformed and logically fortified the sentence. Per reviewer’s preference, in the last round, we eliminated the buffalo and other species from Fig. 2; we have now updated this figure to eliminate any references to the said species. All now refers to small ruminants, cattle, and camel only, in agreement with the rephrased sentence in Sec. 2.1.

- Section 2.2. Network construction.

o 2.2.1. Aggregation models. The reader does not need a textbook on aggregation models, but a simple explanation that you have chosen an aggregation model that considers frequency of movements and headcounts as weights. If the unital model is trivial, then you can skip it. Keep only the relevant information that make the methods reproducible and justified. How are the pairs ordered? In this section it seems pairs are only made of town-town. This should be made clear, i.e. a sum of the number of farms and slaughterhouses was obtained by town and the network constructed between towns of origin and destination.

Response:

  • Per reviewer’s preference, we have made this section shorter, eliminating the description of the unital model.
  • Trivial, as (was) attributed to the unital model, is a technical term, indicative of (only) relative lack of structure.
  • No agency orders the pairs, and both orders of any two given nodes, be it towns, dogs, or stars, are considered when forming a directed network: (town1, town2) and (town2, town1). In the event that, for a particular order of a particular choice of a pair (of nodes), the attributed weight vanishes, this particular ordered pair is absent from the resulting directed network. As the topic is textbook standard, we did not expand.
  • Although we feel that the construction schemes for frequency and volume models are self-sufficiently explained where they are introduced, they were only meant to complement what can already be found in the cited work, (Volkova, Howey et al. 2010), from which the models are borrowed.

o 2.2.2. Dissimilarity inference. There is a lot of theory here and it is not clear what was applied, why and how it is interpreted or applied to your research. Why do you need to infer dissimilarity, with what purpose? Such explanations are missing

Response: As the section explains, matrix-based metrics require the (degree) of similarity between nodes as their input edge weights in order to produce meaningful results. On the other hand, distance-based metrics require dissimilarities as edge weights. The topic is textbook standard and, by nature, defies theorization: one is constructing something (the network) by subjectively interpreting certain quantifiable relations (resulting in weighted edges) among a set of entities (nodes). So theory is inherently absent while freedom (of interpretation) abounds; which is why we expand on the topic to the extent we are properly equipped to describe the choices we made.

o 2.2.3. Orientation reversal. The same comments to 2.2.2 apply here regarding why was it applied, what is being done, why and how it is interpreted.

Response: As was indicated in the revised manuscript, the reversal is done to translate the meaning of edges from that of supply, as it stands in the network(s) of shipments studied here, to that of demand, as it is used in computing hub and authority scores where edges are presumed to encode referral. We expanded on the subject, explaining why the reversal is done in general and what it entails in the particular case of hub and authority scores that are later used in our hybrid centrality constructions.

- Section 2.3. Static measures.

o 2.3.1. Global clustering coefficient. Please explain what the first two sentences mean and rewrite more clearly. What is Tr in formula (1)? How is this coefficient interpreted? Why are you applying it, with what purpose?

Response:

  • A triangle is a closed path of length three.
  • A quantity is normalized in order to scale it wrt a certain characteristic of the network, of which the scaled/former quantity is a special case; in this case the latter is the number of paths of length two. Typically in textbooks, where the global clustering coefficient is introduced, this is sufficiently explained.
  • Transitivity is a basic mathematical property of a relation (on a set (of objects)); it states that, for every three objects A, B, and C from the set, if (A, B) and (B, C) are (ordered) pairs of the relation, so is (A, C). In the case of a (directed) network, the (directed) relation defines its (directed) edges. This is textbook standard and, as its (partial) presence implies the existence of cycles of minimal length (three), it can be used as a (partial) marker of deviation from hierarchical arrangements (as network structure).
  • Tr is standard matrix notation; it stands for the trace operator, as is now also explained in the text.
  • Network measures quantify different characteristics of the network. In time slice analysis, one ideally looks for a subset of those measures whose sliced behavior in time can be used as markers of eras of phenomenological interest. In the particular case of global clustering coefficient we did not, given our current understanding of the temporal aspects of the shipment networks in the given timespan of the data, recognize a distinct era; however a subjective null result is a result in its own and we have chosen to include it in our study.
  • The formula is included as there exist different formulations of weighted global clustering and it is necessary to specify which formulation was used.

o 2.3.2. Global efficiency- the only bit missing here is why is this measure important

Response: The global notion of efficiency is used to demonstrate that of forward and backward efficiencies; the latter are not commonly discussed but we use them as starting points to further construct our source and sink centralities, and eventually our hybrid versions of the latter two that we solely employ in Results.

o 2.3.3. Local efficiencies. Why is this restriction needed in addition to global efficiency? Did you fix the first index, the second index or both? How do you fix the indeces? What is the added value of measuring the forward and backward efficiencies?

Response: This is partly answered in the previous item. Fixing the indices means to eliminate the summation on that index which renders the index a variable, indicating the node it represents. So the local efficiencies are computed wrt a given node, and not for the entire network. As , the distance from node i to node j, is asymmetric, fixing the first index i computes a different quantity wrt node i from that of fixing the second index j wrt node j; these are (local/node-wise) forward and backward efficiencies, respectively.

- Section 2.4. Hub and authority scores. Here the new text incorporates many of the suggestion I have raised for previous sections. It is not clear however why you need to reweight your modified forward and efficiency scores. In the authors’ response, this has been named hybrid and sink scores. Please clarify.

Response: Please see the discussion starting on line 301; it is there to perform this very task.

- Section 2.5. Hybrid centralities. Can you please summarize which centrality measures are you considering and why. How do you rank the nodes to obtain a and b.

Response: This is a composition template, given any two centrality measures a and b, with a estimated primary and b secondary. It is used in the construction of source- and sickness centralities, as well as in the transition from the latter to their hybrid counterparts. In the second case a represents the source-/sinkness centrality of a node, while b that of hub/authority score (centrality).

- Section 2.6. Prevention versus detection. Please clarify what a sink is referring to in the text. Please also clarify and summarize (i.e with a table) which measures have you chosen for your study, why and how they are interpreted of those you mention in a theoretical way: betweenness, weighted outdegree (clarify what are the out-edges of the node and its unweighted outdegree), shortest path, inverse distance, sourceness.

Response:

  • The classic notions of source/sink are defined as nodes of zero in-/out-degree and, so long as the node is not isolated, inevitably nonzero out-/in-degree.
  • The notion of an out-/in-edge of a node is textbook standard; it refers to an edge whose direction points away/toward the given node. In terms of ordered relations, it is an element of the relation that defines the edge set of the network with the given node as its first/second element.
  • The out-/in-degree of a node is the sum of the weights of its out-/in-edges; this is textbook standard. In computing their unweighted versions, just as in the unital aggregation model, the weights are perceived unital in their respective sums; so, in effect, they quantify the number of such edges.
  • The only centrality measures that we use in this study are the hybrid versions of source- and sickness centralities that we incrementally build up to in this section. Betweenness is only used for comparison with weighted outdegree which, in turn, is but one ingredient in the construction of the sourceness score. The shortest path between the ordered pair of nodes i and j is what stands for, their generally asymmetric distance; this is textbook standard. Inverse distance refers to the inverse of the latter quantity (as in “one over..”).

- Section 2.7. Temporal analysis. It is not clear why of the three common approaches the first was chosen and why, and how it is interpreted. While it is of great information to know that there are three common approaches, the text could be narrowed and simplified to explain better the method chosen, rather than introducing every method available.

Response:

  • Given its rudimentary nature, we feel that the chosen approach is adequately described in the first paragraph. And precisely because of the fact that our method of choice is rudimentary, we acknowledge the other two approaches that we fell short of investigating. The text is updated to reflect our compromise.
  • Only overall strategies (approaches) are enumerated, and not individual methods, resulting in the frugal collapse of every to three.

- Section 2.8. Spatial analysis. Which aggregation method was used?

Response: The use of the term is literal and clarification was made in the text.

RESULTS.

- Figure 2. “All” here includes also buffalo and other species, as stated in the legend, or without them, as stated in M&M? It is also still not clear the interpretation of the graph. In the example in the new legend, 29,173 are directed to farms, but the number appears in the light green part of the column, which corresponds to movements to slaughterhouses. Please clarify/correct as necessary.

Response: The figure and its caption are updated. All is now referring to small ruminants, cattle, and camel only.

- The sentence in lines 346-348 is needed in M&M, not here.

Response:

- Please summarize the towns which have the greatest volume/frequency in Figure 3 in the text, and explain where they are located (province)

Response:

It has been added (in lines 401-411). Thanks.

- Please rearrange either M&M sections or Results so that sections under both epigraphs match

Response: Unfortunately this is not feasible since

  • some of the topics under “2.2 Network construction” are pertinent to all networks while some apply only to centrality estimates; moreover,
  • some of the topics under “2.3 Static measures” pertain to centrality estimates while one is specific to temporal analysis.

- Section 3.1. Temporal description. Please remind the reader of the timespan of the dataset. None of this has been presented in methods- please include in methods that you are going to describe the temporal evolution in regards of Tabaski and Nowruz, for example. Please rearrange M&M sections to match those of the Results.

Response:

- Section 3.2. Spatial description. You must state in the M&M sections that you will summarize your results as strip plots, that you will stratify your results by species or that you will compare importing/exporting towns-none of this has been stated in M&M

Response:

It has been amended (in lines: 327-341). Thanks.

Moreover, in the strip plots (section “a” in the figures No.7-12) we have indicated the statistical indexes of volume of moved animals, and we believe that describing these metrics in the text would make the manuscript less reader-friendly. So, we didn’t repeat “min, max, median, mean” metrics in the text.

o Town centrality estimates. Only hybrid source and sinkness scores are reported- again, an indication that the M&M needs further rearrangements to reflect the results

Response: Please see the explanation starting on line 296; only hybrid source- and sickness centralities were, and are, built toward in M&M.

o Lateral trajectories of records- where is this in M&M?

Response:

A paragraph has been added.

Comments on the Quality of English Language

The English is fine.

Submission Date

14 July 2023

Date of this review

7 Sep 2023 14:56:45

Round 3

Reviewer 2 Report

I have no more comments